# NG2 glia are required for vessel network formation during embryonic development

**Shilpi Minocha[1][†], Delphine Valloton[1][†], Isabelle Brunet[2], Anne Eichmann[2], Jean-Pierre Hornung[1][‡], Cecile Lebrand[1]\*[‡]**

[1]Department of Fundamental Neurosciences, University of Lausanne, Lausanne, Switzerland; [2]INSERM U1050, Collège de France, Paris, France

**Abstract** The NG2[+] glia, also known as polydendrocytes or oligodendrocyte precursor cells, represent a new entity among glial cell populations in the central nervous system. However, the complete repertoire of their roles is not yet identified. The embryonic NG2[+] glia originate from the Nkx2.1[+] progenitors of the ventral telencephalon. Our analysis unravels that, beginning from E12.5 until E16.5, the NG2[+] glia populate the entire dorsal telencephalon. Interestingly, their appearance temporally coincides with the establishment of blood vessel network in the embryonic brain. NG2[+] glia are closely apposed to developing cerebral vessels by being either positioned at the sprouting tip cells or tethered along the vessel walls. Absence of NG2[+] glia drastically affects the vascular development leading to severe reduction of ramifications and connections by E18.5. By revealing a novel and fundamental role for NG2[+] glia, our study brings new perspectives to mechanisms underlying proper vessels network formation in embryonic brains.

**\*For correspondence:** cecile.lebrand@unil.ch

[†]These authors also contributed equally to this work
[‡]These authors contributed equally to this work

**Competing interests:** The authors declare that no competing interests exist.

## Introduction

The vascular system consists of two highly organized, branched, and stereotypic circuits of blood vessels and lymphatic vessels (*Adams and Alitalo, 2007*; *Larrivee et al., 2009*). Blood vessels develop by two unique mechanisms: vasculogenesis, de novo synthesis involving differentiation of cells from endothelial precursor cells (angioblasts), and angiogenesis, formation of vessels from pre-existing vessels by either (a) sprouting, proliferation, and migration of endothelial cells (ECs) or (b) bifurcation of a preexisting blood vessel into two (*Adams and Alitalo, 2007*; *Jain, 2003*; *Larrivee et al., 2009*).

The selection of sprouting ECs is controlled positively by the pro-angiogenic vascular endothelial growth factor (VEGF) and negatively by pericytes, extracellular matrix molecules or VEGF inhibitors. Thereafter, the endothelial tip cells have the capacity to extend filopodia in order to respond to attractive or repulsive signals of the environment. Guidepost cells have been shown to direct tip cells pathfinding by cell contact adhesion or by secreting soluble guidance cues. Therefore, endothelial tip cells share numerous similarities with axonal growth cones and it is admitted that axons and blood vessels use similar mechanisms in order to form complex networks (*Carmeliet and Tessier-Lavigne, 2005*; *Eichmann et al., 2005*). Additionally, pathfinding of blood vessels has been shown to require Sema3E/PlexinD1, Netrin-1/Unc5B, and Slit/Robo4 signaling (*Adams and Alitalo, 2007*; *Carmeliet and Tessier-Lavigne, 2005*; *Lu et al., 2004*; *Larrivee et al., 2007*; *Eichmann et al., 2005*). Sprout extension is made by the migration of the stalk ECs behind the tip cell or by local proliferation of stalk ECs. The fusion of adjacent sprouts and vessels that occurs after tip cells encounter each other is regulated by adhesive or repulsive interactions.

Blood vessels in the telencephalon of embryonic mice are divided into pial or periventricular vessels based on their anatomical location, their growth patterns and development. The periventricular vessels are believed to give birth to the arterial networks, while the pial vessels may generate the

**eLife digest** In the brain, nerve cells and blood vessels form complex networks that are interconnected. The networks of blood vessels form as the main regions of the brain develop in the embryo. During this time, glial cells provide physical support to the developing nerve cells and produce signals that guide them to the correct location. NG2[+] glial cells are different to other types of glial cells. They spread throughout the developing brain and form complex, highly branched shapes. They also appear to be involved in several other processes in the developing brain as they give rise to other glial cells and to neurons. However, the full scope of their roles during brain development was not clear.

Here, Minocha, Valloton et al. investigate the role of NG2[+] glia in the developing mouse brain by first marking the cells that form NG2[+] glia with a fluorescent protein. This approach allowed Minocha, Valloton et al. to track where and when NG2[+] glia form. The experiments revealed that midway through the development of mouse embryos, these glia populate an entire region called the telencephalon, which later forms a region of the brain called the cerebral cortex. The appearance of these cells coincides with the formation of the brain's blood vessel networks. These early NG2[+] glia cells then disappear a few days after birth.

Minocha, Valloton et al. also found that NG2[+] glia form many close contacts with blood vessels in the cortex. The cells are found at the tip of newly branching vessels, or are tethered along the walls of the vessels. Furthermore, the NG2[+] glial cells also create bridges between neighboring vessels. When these glia are missing from the brains of mouse embryos, blood vessel networks in the telencephalon fail to branch normally and connect with each other. Thus, NG2[+] glial cells are important for the branching, connection and refinement of blood vessels in the cerebral cortex during development. Future studies will aim to identify which of the molecules produced by glial cells influence the formation of the blood vessel network.

venous sinuses (*Hiruma et al., 2002*). As soon as E9, developing pial vessels forming a vascular plexus surround the entire brain without following any obvious spatial or temporal gradient. By contrast, the periventricular vessels develop in a ventral-to-dorsal gradient within the telencephalon. Angiogenesis in the embryonic telencephalon, in addition to the extrinsic factors listed above, is also controlled by a set of region-specific transcription factors, from the ventral telencephalon such as Nkx2.1 and Dlx2, and from the dorsal telencephalon like Pax6 that are expressed in a subset of ECs (*Vasudevan and Bhide, 2008*; *Vasudevan et al., 2008*). The distinct spatial and temporal expression of these transcription factors has been shown to direct telencephalic vascular development.

Until now, macrophages that lie in close proximity to the blood vessels have been known to act as angiogenic agents and have been implicated in blood vessel development during growth and repair (*Fantin et al., 2010*; *Newman and Hughes, 2012*; *Nucera et al., 2011*; *Outtz et al., 2011*; *Pollard, 2009*). In brain, early embryonic macrophages travel from the yolk sac, express Tie2 and the neuropilin 1 receptor, and are present at the time of brain vascularization. They were observed to localize at vessels junctions and interact with the endothelial tip cells, by forming bridges to align them and to prepare them for the later fusion (*Fantin et al., 2010*; *Nucera et al., 2011*; *Outtz et al., 2011*). In this paper, we discovered that another class of glial cells, NG2[+] glia, are also involved in the vascular network formation in the early mouse telencephalon.

NG2[+] glia or polydendrocytes constitute a population of cells that are different from neurons, mature oligodendrocytes, astrocytes, and microglia (*Nishiyama et al., 2002*; *Nishiyama et al., 2009*). They express NG2 (Nerve/glial antigen 2 or CSPG for chondroitin sulfate proteoglycan) and Olig2 (Oligodendrocyte precursor bHLH transcription factor 2) but lack astrocytic markers namely GFAP (glial fibrillary acidic protein) and GLAST (Astrocyte-specific glutamate and aspartate transporter). They display complex highly branched morphology and are uniformly distributed within the grey and white matter throughout all layers. They generate oligodendrocytes in vitro and have been considered since a long time as oligodendrocyte progenitor cells (*Polito and Reynolds, 2005*; *Nishiyama et al., 1996a*, *1996b*, *2009*; *Zhu et al., 2008b*). However, recently, NG2[+] glia have also been shown to differentiate into neurons and protoplasmic astrocytes in the grey matter as well as

in the white matter (*Rivers et al., 2008*; *Zhu et al., 2008b*; *Zhu et al., 2008a*). Furthermore, electro-physiological studies indicate that NG2$^+$ glia receive synaptic input from neurons, and like astroglia, probably participate to the neuronal network (*Paukert and Bergles, 2006*; *Butt et al., 2005*). NG2$^+$ glia have also been implicated in synaptic reorganization after cortical and spinal cord injury, but their precise role is still unknown (*Nishiyama, 2007*; *Nishiyama et al., 2009*).

This study displays a novel function for this rather recently classified glial cell population. The embryonic NG2$^+$ glia of the dorsal telencephalon originate from the subpallial Nkx2.1$^+$ progenitors. Interestingly, they occupy the telencephalon at the same time as the establishment of the blood vessel network. Additionally, they not only originate at the same time but also reside very closely to the developing vessel network. Hence, the strategic temporal occurrence is conjugated with spatial proximity to the vessel network. They are closely juxtaposed to the blood vessels, either at sprouting tip cells, branching points or along the vessel walls. Remarkably, in their absence, the embryonic vasculature was poorly developed and exhibited reduced connectivity. Our results bring forth a new function for this class of glia. In summary, our findings propose the participation of *Nkx2.1*-derived NG2$^+$ glia toward blood vessel network formation and stabilization during late embryonic ages. Therefore, this study gives new insights into the mechanisms involved in brain angiogenesis and implies that transient NG2$^+$ glia work together with macrophages in guiding vessels.

## Results

### Identification of transient *Nkx2.1*-derived NG2$^+$ glia in embryonic telencephalon

Here, we identified early NG2$^+$ glia within the embryonic telencephalon and investigated their function during development. To this purpose, we first characterized the molecular identity and origin of the embryonic NG2$^+$ glia that populated the telencephalon. To selectively fate map the NG2$^+$ glia, we used NG2 immunostaining in wild-type (WT) mice, *Cspg4-cre$^+$/Rosa-EYFP* (CSPG or chondroitin sulfate proteoglycan also known as NG2) and *Nkx2.1-cre$^+$/Rosa-EYFP* mice. Between E12.5-to-E16.5, we observed that the NG2$^+$ glia were positioned within the marginal zone (MZ), the subplate, the intermediate zone (IZ) and the sub-ventricular zone (SVZ) of the lateral cortical area, and in the septum (SEP) of the WT mice (*Figure 1*). They populated the midline corticoseptal boundary (CSB) region at E12.5 (n=3) (*Figure 1A*), and the cingulate bundle (CI), the cingulate (CCi) and frontal (CFr) cortices at E14.5 (n=3) (*Figure 1B and 1C*). By E16.5, NG2$^+$ glia were ubiquitously dispersed within the WT dorsal telencephalon (n=3) (*Figure 1D*).

We then analyzed in detail the *Cspg4-cre$^+$/Rosa-EYFP* mice wherein the NG2 promoter dictated specific Cre recombinase expression which then lead to permanent YFP expression from the constitutively active Rosa promoter. In *Cspg4-cre$^+$/Rosa-EYFP* mice, the YFP signal was detected in a majority of embryonic NG2$^+$ glia of the dorsal telencephalon (at E18.5: 71.7 ± 14.6% in the corpus callosum (CC), 57 ± 3.8% in the CI and 69 ± 5.3% in the CCi; n=3) (*Figure 2—figure supplement 1A*). The entire cell population visualized by the YFP signal at E16.5–E18.5 co-expressed NG2 (n=3) and Olig2 (n=3), two well-known markers for NG2 glia, and also S100β (n=3) , considered as a marker for astrocytes and NG2$^+$ glia (*Cahoy et al., 2008*; *Honsa et al., 2012*; *Rivers et al., 2008*) (*Figure 2A–C* and *Figure 2—figure supplement 1E*). As expected, at same ages they did not express the specific astrocytic markers GLAST (n=3) and GFAP (n=3) (*Figure 2D–E* and *Figure 2—figure supplement 1E*). Although immunostaining showed that in WT mice, PDGFR-β$^+$ pericytes adjacent to the vessels were NG2$^+$ (*Figure 3D*), Cre-mediated recombination in *Cspg4-cre$^+$/Rosa-EYFP* mice did not occur properly in the pericytes. As a result, although NG2 is expressed by pericytes (*Levine and Nishiyama, 1996*; *Stallcup and Huang, 2008*; *Virgintino et al., 2007*), we found only very few PDGFR-β$^+$ pericytes labeled for the YFP in *Cspg4-cre$^+$/Rosa-EYFP* telencephalon (*Figure 3B*, n=3). A substantial proportion of the PDGFR-β$^+$ pericytes population was YFP$^-$. Quantifications of the two populations: PDGFR-β$^+$/YFP$^-$ pericytes and PDGFR-β$^+$/YFP$^+$ pericytes showed that only 4.95 ± 1.54% of total PDGFR-β$^+$ pericyte-population was co-labeled with YFP (*Figure 3G*, n=10). Thus, vast majority of the YFP signal in *Cspg4-cre$^+$/Rosa-EYFP* brains was present in NG2$^+$ glia alone.

As *Nkx2.1*-regulated precursors have been previously shown in embryos to produce transient oligodendrocyte precursor cells (OPCs) in addition to giving rise to GABAergic interneurons and

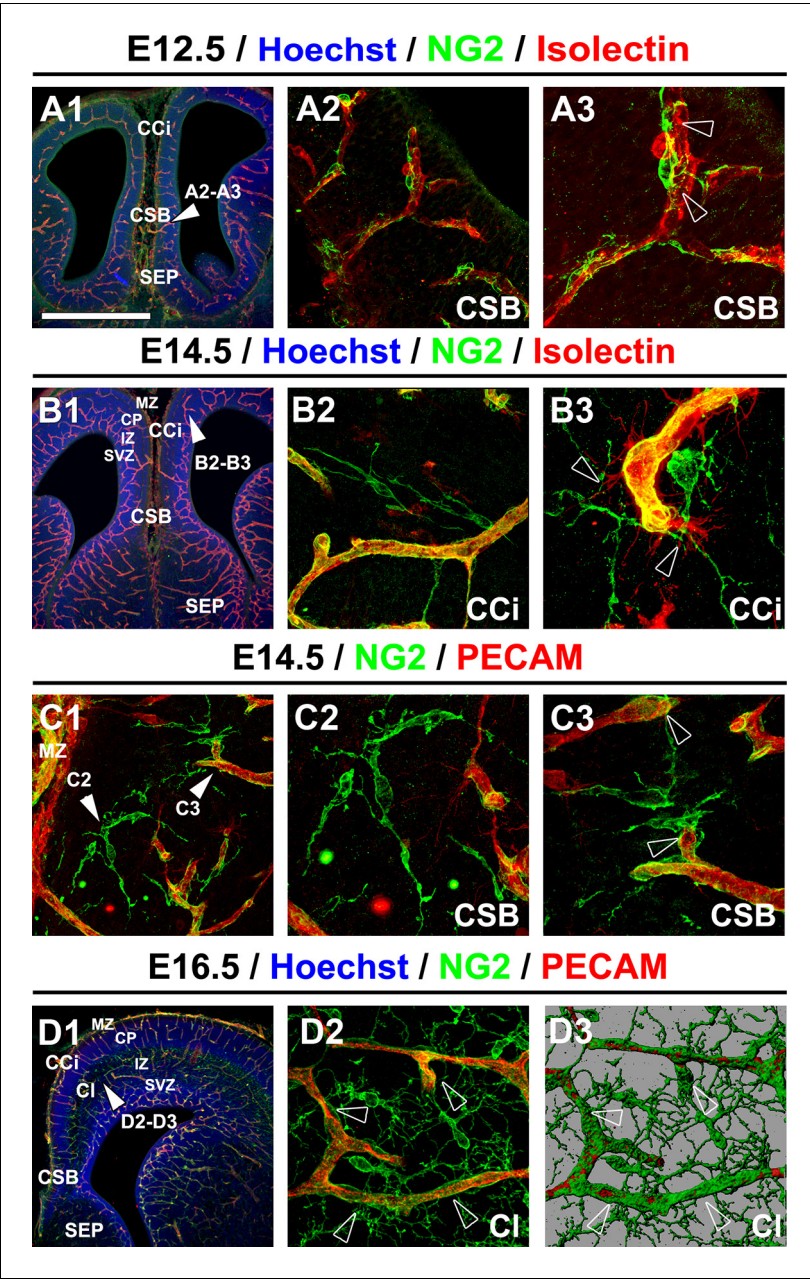

**Figure 1.** NG2[+] glia are in close contact with blood vessels. (**A–D**) Double immunohistochemistry for NG2 and Isolectin (**A1–A3, B1–B3**) and for NG2 and PECAM (**C1–C3, D1–D2**) on coronal cingulate cortex (CCi) and cingulate bundle (CI) sections of wild-type mice (n=3 each) at E12.5 (**A1–A3**), at E14.5 (**B1–B3** and **C1–C3**), and at E16.5 (**D1–D2**). **A3, A2, B2, B3, C2, C3,** and **D2** are higher power views of the region in **A1, B1, C1,** and **D1,** respectively (white arrowheads). **D3** is an isosurface reconstruction of the labeling seen in **D2**. The processes of the NG2[+] glia are in close contact with adjacent blood vessels (open arrowheads in **A3, B3,** and **C3**). Bar = 675 μm in **A1, B1,** and **D1**; 50 μm in **A2, B2, C1,** and **D2**; 40 μm in **A3, B3, C2,** and **C3**. CSB, corticoseptal boundary at the midline where the corpus callosum will form.

astrocytes (*Nery et al., 2001*; *Kessaris et al., 2006*, *Minocha et al., 2015*), we decided to make use of the *Nkx2.1-cre[+]/Rosa-EYFP* mice to look further at the subpallial origin, time of appearance and spatial arrangement of the embryonic NG2[+] glia that ubiquitously occupied the dorsal telencephalon toward the end of embryonic development. Using our *Nkx2.1-cre[+]/Rosa-EYFP* mice, our findings confirmed that *Nkx2.1*-derived precursors produced YFP[+]/NG2[+] glia that colonized the cingulate

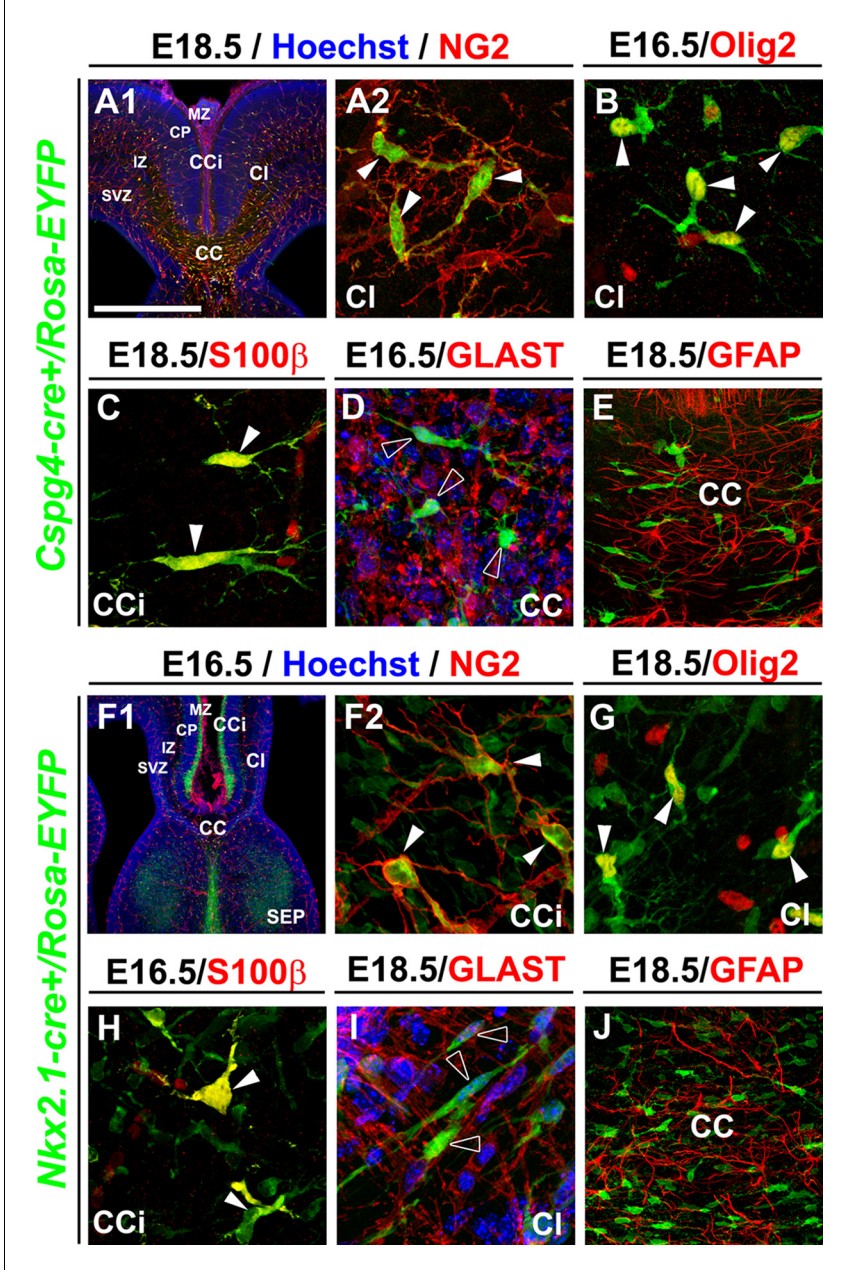

**Figure 2.** NG2[+] glia of the dorsal telencephalon are derived from Nkx2.1[+] progenitors of the subpallium. (A–E) Double immunohistochemistry for the YFP and NG2 (A1–A2) (n=3), the YFP and Olig2 (B) (n=3), the YFP and S100β (C) (n=3), the YFP and GLAST (D) (n=3), and the YFP and GFAP (E) (n=3) on telencephalic coronal slices of *Cspg4-cre[+]/Rosa-EYFP* mice at E16.5 (B and D) and E18.5 (A1, A2, C, and E). (F–J) Double immunohistochemistry for the YFP and NG2 (F1–F2) (n=5), the YFP and Olig2 (G) (n=5), the YFP and S100β (H) (n=4), the YFP and GLAST (I) (n=4), and the YFP and GFAP (J) (n=3) on telencephalic coronal slices of *Nkx2.1-cre[+]/Rosa-EYFP* mice at E16.5 (F1, F2, and H) and E18.5 (G, I, and J). A2 and F2 are higher power views of the cingulate region in A1 and F1, respectively. The *NG2*-derived and the *Nkx2.*1-derived YFP[+] cells co-express polydendroglial markers, NG2 and Olig2, together with S100β (white arrowheads) but lack expression of GLAST and GFAP (open arrowheads in D and I). Bar=675 μm in A1, F1; 100 μm in E, J; 50 μm in A2, B, C, D, F2, G, H, I.

The following figure supplements are available for figure 2:

**Figure supplement 1.** YFP signal in *Nkx2.1-cre[+]/Rosa-EYFP* and *Cspg4-cre[+]/Rosa-EYFP* mice is present in NG2 glia.

**Figure supplement 2.** *Nkx2.1*-derived NG2 and Olig2 glia are transient and gradually disappear from the dorsal pallium at postnatal ages.

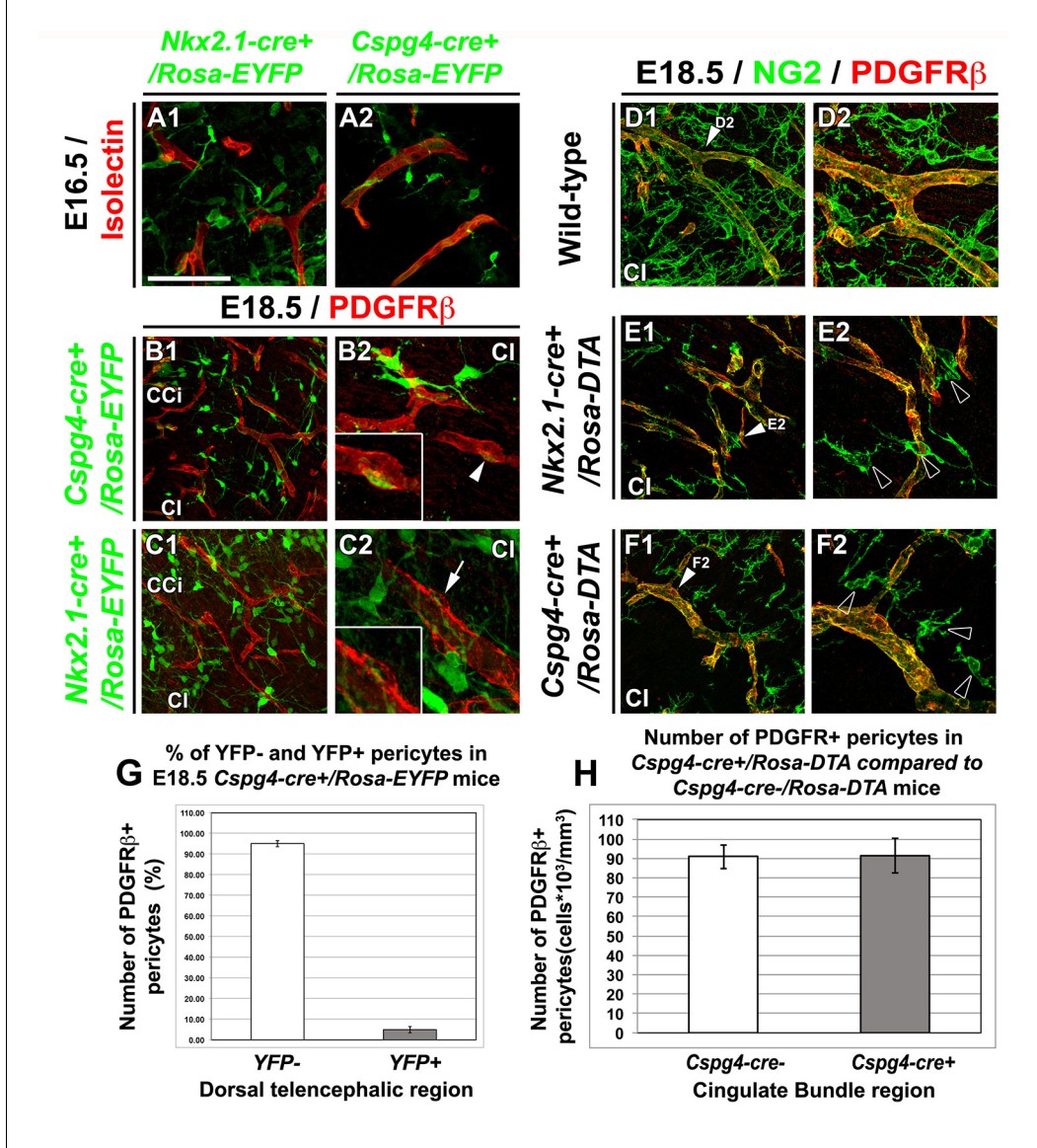

**Figure 3.** NG2[+] glia, but not pericytes, control blood vessels formation. (**A–C**) Double immunohistochemistry for the YFP and Isolectin (**A1–A2**) or PDGFRβ (**B1–B2, C1–C2**) on coronal cingulate cortex (CCi) and cingulate bundle (CI) sections of *Nkx2.1-cre+/Rosa-EYFP* (**A1, C1–C2**) and *Cspg4-cre+/Rosa-EYFP* (**A2, B1–B2**) mice (n=3) at E16.5 (**A1–A2**) and E18.5 (**B1–B2** and **C1–C2**). (**A–C**) From E16.5 to E18.5, numerous YFP[+] NG2 glia are surrounding cortical blood vessels. (**B–C**) The Cre-mediated recombination, visualized by the YFP signal, can be observed in only very few pericytes surrounding blood vessels in *Cspg4-cre+/Rosa-EYFP* mice (**B2**, boxed region showing high magnification of region marked with white arrowhead), but not in *Nkx2.1-cre+/Rosa-EYFP* mice (**C2**, boxed region showing high magnification of region marked with arrow). (**D–F**) Double immunohistochemistry for NG2 and PDGFRβ on coronal CI sections in wild-type (**D1–D2**), *Nkx2.1-cre+/Rosa-DTA* (**E1–E2**) (n=3) and *Cspg4-cre+/Rosa-DTA* (**F1–F2**) (n=3) mice at E18.5. (**D–F**) The NG2[+] glia form a complex cellular network around the cortico-cerebral blood vessels outlined by NG2 and PDGFRβ staining. The DTA under the control of *Nkx2.1* (**E**) and *Cspg* or *NG2* (**F**) promoters selectively depletes NG2[+] glia but not pericytes. **D2, E2,** and **F2** are higher power views of the cingulate region in **D1, E1,** and **F1**, respectively (white arrowheads). (**G**) Bars (means ± SEM) represent the percentage of YFP-negative and YFP-positive PDGFRβ labeled pericytes in dorsal telencephalon sections of E18.5 *Cspg4-cre+/Rosa-EYFP* mice (n=10). The YFP signal in *Cspg4-cre+/Rosa-EYFP* mice was not detected in PDGFRβ[+] embryonic pericytes of the dorsal telencephalon (95.05 ± 1.54% of pericytes are YFP-negative in the CI at E18.5, n=10). (**H**) Bars (means ± SEM; unpaired Student's *t*-test) represent the percentage of remaining PDGFRβ+ pericytes in cingulate bundle (CI) sections of E18.5 *Cspg4-cre+/Rosa-DTA* mice (n=11) compared to control mice

*Figure 3 continued on next page*

*Figure 3 continued*

(n=11). No loss of PDGFRβ+ pericytes was observed in *Cspg4-cre*[+]/*Rosa-DTA* mice compared to *Cspg4-cre*[-]/*Rosa-DTA*. Bar = 100 μm in **B1, C1**; 50 μm in **A1, A2, D1, E1, F1**; 40 μmin **B2, C2, D2, E2, F2**.
The following figure supplements are available for figure 3:

**Figure supplement 1.** Drastic depletion of GAD67-GFP[+] neurons in *Nkx2.1*[-/-], *Nkx2.1-cre*[+]/*Rosa-DTA* and *Nkx2.1-cre*[+]/*Eno2-DTA* cortices.

**Figure supplement 2.** GLAST[+] astrocytes are not affected in *Nkx2.1-cre*[+]/*Rosa-DTA* cingulate cortex at E18.5.

cortical area and the midline (*Figure 2F*, *Figure 2—figure supplement 1F*) (*Kessaris et al., 2006*). The YFP signal was detected in all embryonic NG2[+] glia of the dorsal telencephalon from E16.5 to E18.5 (100% colocalization in the CC, and the CCi at E16.5 and E18.5; n=5) (*Figure 2F*). *Nkx2.1*-derived NG2[+] glia expressed Olig2 (n=5) and S100β (n=4) (*Figure 2G–H*), and lacked GLAST (n=4) or GFAP (n=3) expression (*Figure 2I–J*). This further emphasized that subpallial domains are sites for early NG2[+] glia genesis. Furthermore, we followed the presence of these YFP[+] /NG2[+] glia after birth. We observed that the YFP[+]/Olig2[+]/NG2[+] glia originating from Nkx2.1 domains were transient in nature and disappeared abruptly from the cortex around P8 (n=3) (*Figure 2—figure supplement 2*). These results are coherent with previous studies, wherein the OPCs generated from the Nkx2.1-expressing precursors were shown to disappear around P10 (*Kessaris et al., 2006*). Interestingly, we never detected any YFP signal in the ECs and the pericytes of the cerebral vasculature (*Figure 3A1 and 3C* ; n=5). Thus, based on the specificity of the Cre- reporter strains (both *Cspg4-cre* and *Nkx2.1-cre*) for NG2[+] glia, these mice can be further utilized to determine the function of the embryonic NG2[+] glia independent of ECs and pericytes.

These results altogether show that in embryos, NG2[+] glia of the dorsal telencephalon are derived from Nkx2.1[+] progenitors of the subpallium and transiently occupy the telencephalon. The disappearance of the *Nkx2.1*-derived NG2[+] glia a few days after birth underlines their functional requirement for events occurring during embryonic and early postnatal period.

## Spatial association between embryonic *Nkx2.1*-derived NG2[+] glia and developing vessels

We further studied the precise localization of the embryonic NG2[+] glial population to elucidate their functional relevance. At E14.5 (n=3), the first pioneer NG2[+] glia were observed in the CCi cortical plate (CP) (*Figure 1*). As such, they occupied the CP at the same time as establishment of the blood vessel network and were seen to accompany it while growing and invading the dorsal telencephalon. The NG2[+] glia were present from E14.5-to-E16.5 throughout the dorsal telencephalon during the early phase of angiogenesis when vessels begin to extend, sprout, and form new branches, as well as during the later phase from E16.5-to-E18.5 when branches fuse together or retract (*Figures 1* and *2*). Interestingly, we observed that NG2[+] glia formed a complex cellular network around the cerebral vessels outlined by NG2, PECAM, Isolectin, or PDGFR-β labeling (*Figures 1* and *2*). They were localized at sprouting tip cells or at branch fusion points, and also all along the vessel walls (*Figure 1*). Several long and slender processes of the NG2[+] glia appeared to wrap or tether to the vascular walls, possibly reflective of a functional interaction (*Figure 1A3, 1B3, 1C3*; open arrowheads). The interactions between blood vessel network and NG2[+] glia enhanced as embryonic age progressed (compare *Figure 1A* to *1D* and *3D*). A careful analysis using highly magnified three-dimensional (3D) confocal pictures combined with iso-surface representations revealed that NG2[+] glia make multiple close contacts with different cortical blood vessels and created bridges between neighboring vessels (*Figure 1D3*). The bridges are composed of several NG2[+] glia that make connections via their entangled processes, and connect the neighbor vessels (*Figure 1B–D*). The strategic position of NG2[+] glia between the blood vessels created an elaborate mesh-like structure, and reflected the possibility that NG2[+] glia participated toward the vasculature connectivity (*Figure 1B–D*).

Therefore, the location and timing of appearance of the transient NG2[+] glia raise the possibility that they actively participate in the branching and refinement of the cerebral vasculature.

# Brain angiogenesis defects after ablating *Nkx2.1*-derived post-mitotic cells

The concurrence of angiogenic development, NG2[+] gliogenesis together with the elaborate connectivity between NG2[+] glia and vessels made us wonder if the NG2[+] glia were involved in regulating brain angiogenesis. To test this hypothesis, we used mice expressing a 'floxed' diphtheria toxin gene that allows selective ablation of cells in the whole brain (*Rosa-DTA*) (*Brockschnieder et al., 2006*). By crossing *Nkx2.1-cre*[+] mice with *Rosa26-DTA* mice that expressed the diphtheria toxin under the control of the Nkx2.1 promoter, only Nkx2.1-expressing cells were selectively depleted (*Minocha et al., 2015*). *Nkx2.1-cre*[+]/*Rosa-DTA* mice allowed a selective ablation of *Nkx2.1*-derived post-mitotic cells without affecting Nkx2.1[+] precursors because the diphtheria toxin expression under the control of Nkx2.1 promoter took several days (*Minocha et al., 2015*). As the *Nkx2.1*-regulated precursors are known to give rise to GABAergic interneurons (*Anderson et al., 2001*; *Corbin et al., 2001*; *Marin and Rubenstein, 2001*; *Sussel et al., 1999*; *Xu et al., 2008*) and astrocytes (*Minocha et al., 2015*), we aimed to identify the contribution of GABAergic interneurons and astrocytes in our assays, as they can also get ablated in *Nkx2.1-cre*[+]/*Rosa-DTA* mice. To do so, we incorporated the reporter Gad1-EGFP (Gad1 corresponds to the Gad67 gene) into the *Nkx2.1-cre*[+]/*Rosa-DTA* mice to observe the GFP[+] interneurons. In the dorsal telencephalon of *Nkx2.1-cre*[+]/*Rosa-DTA:Gad1-EGFP* (n=5), *Nkx2.1*-derived post-mitotic GABAergic neurons were depleted severely by ~50% at E18.5 (*Figure 3—figure supplement 1B and 1D*).

Next, we aimed to confirm that the GLAST[+] astrocytes are not affected in *Nkx2.1-cre*[+]/*Rosa-DTA* mice brains at E18.5. In *Nkx2.1-cre*[+]/*Rosa-EYFP* and *Nkx2.1-cre*[+]/*Rosa-tdTomato* mice, recombination can only been seen in GLAST[+] astrocytes of the ventral telencephalon, whereas no recombination can be seen in those that occupy the dorsal telencephalon (*Figure 2I* and not shown). Thus, we did not expect to see an ablation in GLAST[+] astrocytes of the dorsal telencephalon with *Nkx2.1-cre*[+]/*Rosa-DTA* mice brains. Indeed, upon co-immunostaining with anti-GLAST, we found that there was no ablation of GLAST[+] cell population in *Nkx2.1-cre*[+]/*Rosa-DTA* mice (n=4) when compared to control *Nkx2.1-cre*[-]/*Rosa-DTA* mice (n=4) in the cingulate cortex (*Figure 3—figure supplement 2*).

Interestingly, staining for NG2 revealed a nearly complete loss of NG2[+] glia in all medio-dorsal cortical areas in all rostrocaudal sections of *Nkx2.1-cre*[+]/*Rosa-DTA* mice brains compared to control mice brains at E16.5 (n=3) and E18.5 (n=6) (*Figure 4B* and *Figure 4—figure supplement 1*). Thus, this result underlined the presence of only *Nkx2.1*-derived NG2[+] glia in the cortical midline regions specifically. No evident morphological changes were seen in lateral cortical regions where *Nkx2.1*-derived NG2[+] glia intermixed with other NG2[+] glia having different identities and origins (not shown). Also, the ECs and pericytes were not ablated in *Nkx2.1-cre*[+]/*Rosa-DTA*, due to the lack of Cre-mediated recombination in these cells (*Figure 3C and 3E*; n=3). The lack of effective Cre-mediated recombination in ECs and pericytes provided us with a useful tool since the other cells that are involved in vessel network formation were not ablated. Thus, we could specifically assess the effects of a narrower population (i.e. NG2[+] glia).

Remarkably, we observed that vessels stained for vessel markers (NG2, PECAM or Isolectin) formed a poorly developed vascular network in multiple telencephalic regions, such as the septum and the cerebral cortices of *Nkx2.1-cre*[+]/*Rosa-DTA* mice (n=4 for PECAM at E14.5 and E16.5; n=3 for NG2; n=4 for PECAM; n=5 for isolectin at E18.5) compared to WT mice (n=4 for PECAM at E14.5 and E16.5, n=6 for NG2; n=8 for PECAM; n= 5 for isolectin at E18.5) as soon as E16.5 and becoming fully evident by E18.5 in all the rostrocaudal levels (*Figure 4B*, *Figure 5B*, *Figure 6B*, *Figure 5—figure supplement 1D* and *Figure 5—figure supplement 2B*). No defects were visualized prior to E16.5 (*Figure 5—figure supplement 1B*). Cortical vessels of *Nkx2.1-cre*[+]/*Rosa-DTA* mice exhibited a drastic and significant reduction of intersections (nodes), connections and of the density of the vascular network (reduction of nodes: 14.1 ± 4.8% , p<0.05; reduction of branches: 14.7 ± 4.9% , p<0.05; reduction of the total volume of the vascular network: 17.78 ± 4.33% , p<0.05; *Figure 5B,G* and *Table 1*; n=4; unpaired Student's *t*-test). Due to the defects observed in branching pattern and connectivity, the regular vascular pattern was not observed any more in mutants, and vessels formed only isolated units (*Figure 5B* and *Figure 5—figure supplement 1D*). After co-staining for Isolectin and lymphocyte antigen LY-76 marker (Ter119) (*Figure 6*) and DAB staining allowing the visualization of erythrocytes (*Figure 5*), our analyses revealed that, in addition, cortical vessels lost their regular diameter and erythrocytes accumulated at the level of enlarged vessel segments

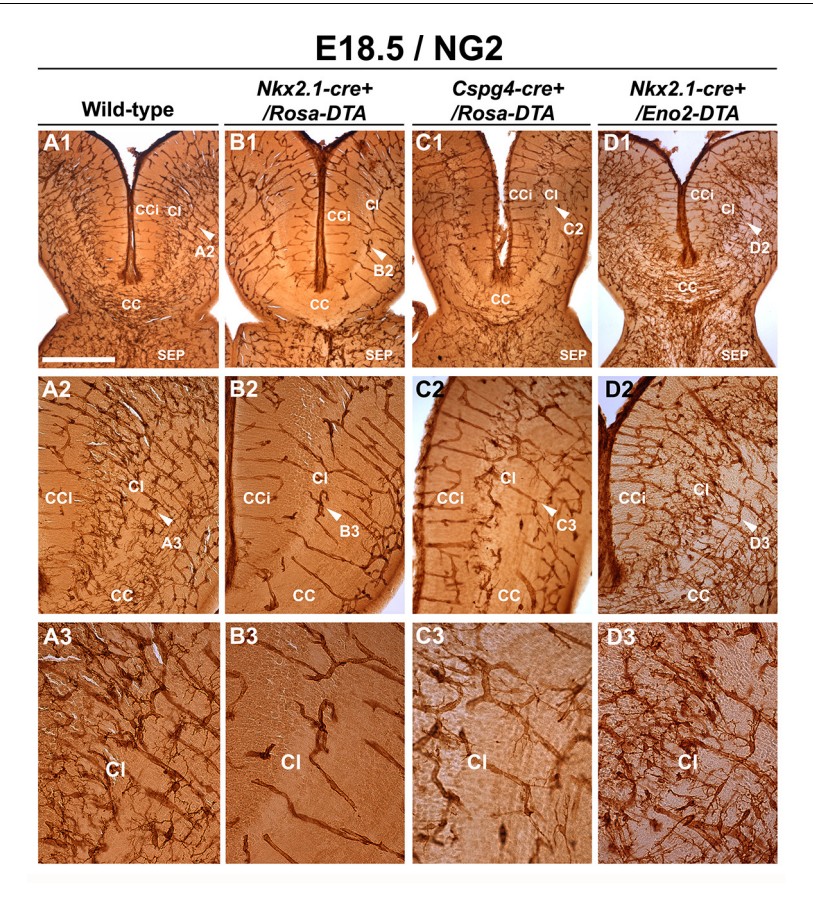

**Figure 4.** Drastic depletion of embryonic NG2$^+$ glia in *Nkx2.1-cre+/Rosa-DTA* and *Cspg4-cre+/Rosa-DTA* midline dorsal telencephalon. DAB staining for NG2 in wild-type (**A1–A3**) (n=6), *Nkx2.1-Cre$^+$/Rosa-DTA* (**B1–B3**) (n=3), *Cspg4-cre$^+$/Rosa-DTA* (**C1–C3**) (n=3), and *Nkx2.1-cre$^+$/Eno2-DTA* (**D1–D3**) (n=3) telencephalic coronal slices at E18.5. NG2$^+$ glia were completely depleted from the corpus callosum (CC), the cingulate cortex (CCi), and cingulate bundle (Cl) of *Nkx2.1-cre$^+$/Rosa-DTA* (**B1–B3**) mutant mice compared to wild-type mice (**A1–A3**). In *Cspg4-cre$^+$/Rosa-DTA* mutant mice (**C1–C3**), there was also a drastic loss of NG2$^+$ cells in medial cortical areas of the dorsal telencephalon with only few remaining cells. In *Nkx2.1-cre$^+$/Eno2-DTA* (**D1–D3**) mutant mice, there was no loss of NG2$^+$ glia in all the observed regions. **A2–A3**, **B2–B3**, **C2–C3**, and **D2–D3** are higher power views of the Cl regions in **A1**, **B1**, **C1**, and **D1**, respectively (white arrowheads). Bar = 500 µm in **A1**, **B1**, **C1**, and **D1**; 250 µm in **A2**, **B2**, **C2**, and **D2**; 125 µm in **A3**, **B3**, **C3**, and **D3**.

The following figure supplement is available for figure 4:

**Figure supplement 1.** Drastic depletion of embryonic NG2$^+$ glia in *Nkx2.1-cre$^+$/Rosa-DTA* midline dorsal telencephalon starts at E16.5.

(*Figure 5E and 5H*, *Figure 6B* and *Table 2*; n=5; unpaired Student's *t*-test). These results are strongly suggestive of brain vessel dysfunction in the mutant mice. Finally, observations made at high magnification after Isolectin and F4/80 staining in both mutant mice did not reveal any defect in tip cell induction and in macrophage recruitment, two processes required for vessels anastomosis (*Adams and Alitalo, 2007*; *Fantin et al., 2010*) (*Figure 6B* and not shown). Quantifications revealed that the number of macrophages was not altered in the *Nkx2.1-cre$^+$/Rosa-DTA* mice compared to control mice, excluding the direct involvement of macrophages in generating the observed defects (not shown).

Next, we tested whether the observed loss of cortical GABAergic interneurons might participate toward the observed vascular defects. In order to address this issue, we selectively ablated neuronal cells without affecting NG2$^+$ glia, by using *Nkx2.1-cre$^+$/Eno2-DTA:Gad1-EGFP* mice. In these mice,

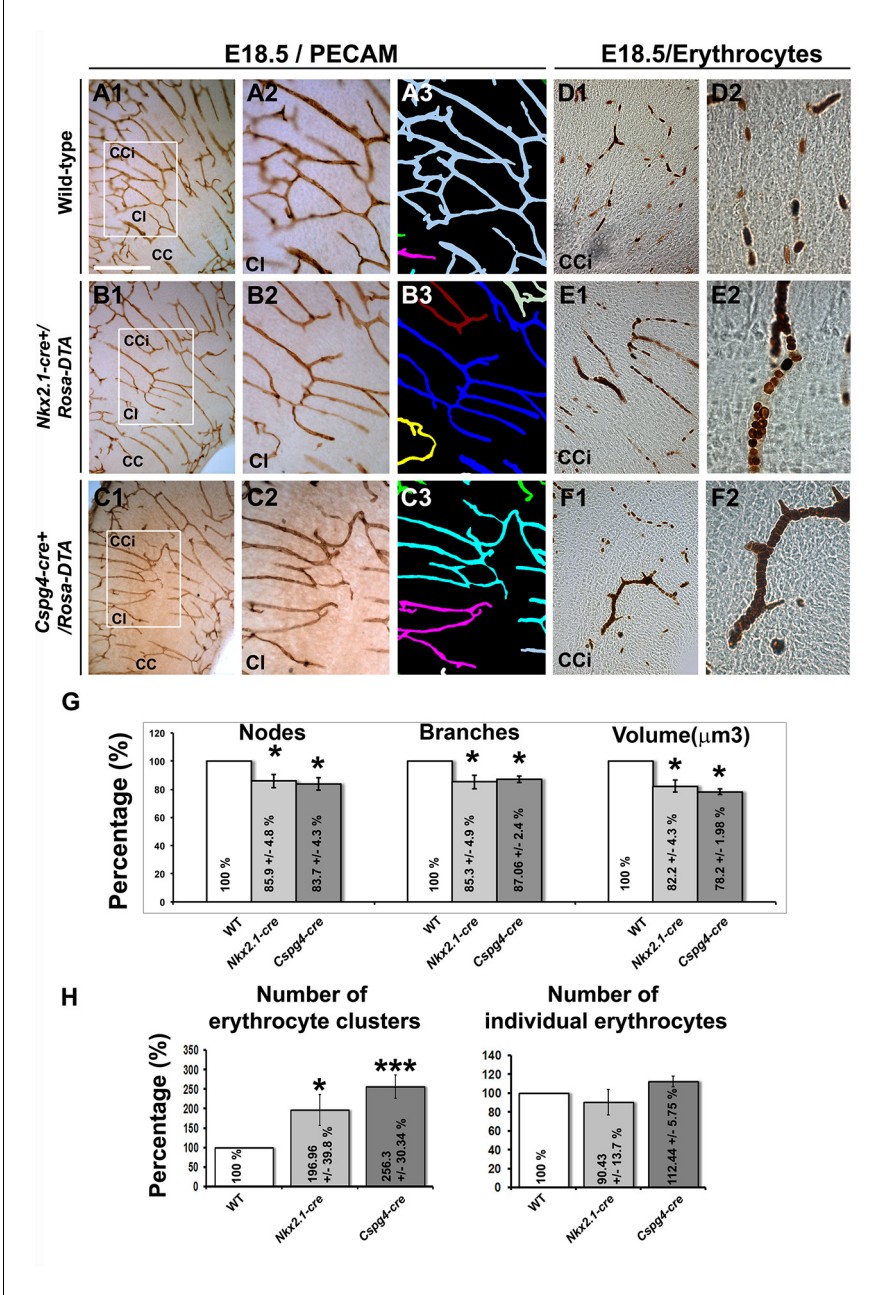

**Figure 5.** Blood vessel branching is similarly impaired in *Nkx2.1Cre+/Rosa-DTA* and *Cspg4-cre+/Rosa-DTA* mice. (A–C) DAB staining for PECAM and reconstitution of the vascular network using the Neurolucida tracing tool in wild-type (**A1–A3**) (n=8), *Nkx2.1-cre+/Rosa-DTA* (**B1–B3**) (n=4) and *Cspg4-cre+/Rosa-DTA* (**C1–C3**) (n=4) cortical coronal sections at E18.5. **A2, B2,** and **C2** are higher magnified views of the boxed regions seen in **A1, B1,** and **C1,** respectively. (D–F) DAB staining for erythrocytes in wild-type (**D1–D2**) (n=10), *Nkx2.1-cre+/Rosa-DTA* (**E1–E2**) (n=5), and *Cspg4-cre+/Rosa-DTA* (**F1–F2**) (n=5) cortical coronal sections at E18.5. (**G**) Bars (means ± SEM) represent the percentages of vessel nodes, vessel branches and volume of the vascular network in mutants compared to wild-type (n=4; unpaired Student's *t*-test). The respective absolute values per CCi section are given in *Table 1*. All the quantified parameters were significantly decreased in mutant mice. (**H**) Bars (means ± SEM) represent the percentage of erythrocyte clusters and individual erythrocytes in mutants compared to wild-type. The respective absolute values per CCi section are given in *Table 2* (n=5; unpaired Student's *t*-test).The number of erythrocyte clusters showed a significant increase in all mutants. The total number of individual erythrocytes remained unchanged. Bar = **A1, B1, C1:** 250 μm; **D1, E1, F1:** 125 μm; **A2, A3, B2, B3, C2, C3:** 62.5 μm; **D2, E2, F2:** 50 μm. CCi, cingulate cortex.

*Figure 5 continued on next page*

*Figure 5 continued*

The following figure supplements are available for figure 5:

**Figure supplement 1.** Embryonic *Nkx2.1*-derived NG2 glia do not control cortical blood vessels outgrowth before E16.5.

**Figure supplement 2.** Blood vessel structure is impaired in the septum of *Nkx2.1-cre+/Rosa-DTA* and *Cspg4-cre+/Rosa-DTA* mice at E18.5.

**Figure supplement 3.** Embryonic *Nkx2.1*-derived GABAergic neurons do not control the development and function of cortical blood vessels.

**Figure supplement 4.** No significant blood vessel network defects are observed in *Cspg4-cre+/Rosa-DTA* postnatal brains.

the diphtheria toxin (DTA) is expressed under the control of a neuron-specific promoter (enolase 2, Eno2) whose action is dictated by the Cre-mediated recombination (Nkx2.1-cre+). In *Nkx2.1-cre+/Eno2-DTA:Gad1-EGFP* mice, we noticed a significant depletion of GAD67-GFP+ GABAergic neurons in the medial cortex (*Figure 3—figure supplement 1C–E*; n=6; unpaired Student's *t*-test) closely equivalent to the loss seen in *Nkx2.1-cre+/Rosa-DTA:Gad1-EGFP* mice. In this mice strain, NG2+ glia were not affected and immunostaining analysis with vessel markers (NG2; n=3 and PECAM; n=4) and erythrocytes (n=3) did not reveal any developmental vessel defects that accompanied the substantial loss of GABAergic neurons in the *Nkx2.1-cre+/Eno2-DTA* cortex (*Figure 4D*, *Figure 5—figure supplement 2* and *3*). Thus, these findings exclude the possibility that the poorly developed vascular network in *Nkx2.1-cre+/Rosa-DTA* mice could be a direct result of GABAergic interneuron loss.

Altogether, these results strongly suggest that the loss of *Nkx2.1*-derived cells, most probably NG2+ glia, was responsible for brain angiogenesis defects.

## NG2+ glia control brain vessels development

To confirm that NG2+ glia play a role in the establishment of the vascular network, we used *Cspg4-cre+/Rosa-DTA* mice in which NG2+ glia were specifically depleted with a drastic loss of 55% ($18.96 \pm 0.98 \times 10^3$/mm³ NG2+ glia in CCi sections of *Cspg4-cre-/Rosa-DTA* mice versus $8.52 \pm 1.06 \times 10^3$/mm³ NG2+ glia in CCi sections of *Cspg4-cre+/Rosa-DTA* mice; n=3, unpaired Student's *t*-test; p<0.001) (*Figure 2—figure supplement 1B–D*, *Figure 3F*, *4C*). Consistent with the very low level of Cre-mediated recombination in the brain vessel pericytes of *Cspg4-cre+/Rosa-EYFP* mice, we did not notice any obvious loss of PDGFR-β+ pericytes in the *Cspg4-cre+/Rosa-DTA* cortex as well (compare *Figure 3F* and *Figure 3D*; n=3). The number of PDGFR-β+ pericytes observed in *Cspg4-cre+/Rosa-DTA* mice (n=11) was similar to those seen in *Cspg4-cre-/Rosa-DTA* mice (n=11) control cingulate bundle, as proved by the quantification analyses (*Figure 3H*). Thus, these results additional confirm that due to the lack of sufficient Cre-mediated recombination in the pericytes, the blood vessel network seen with *Cspg4-cre+/Rosa-DTA* mice brains can be majorly attributed to the NG2+ glia.

The vessel defects in the cerebral cortex and septum of *Cspg4-cre+/Rosa-DTA* mice were seen at all rostrocaudal levels and were similar to those that were aforementioned for *Nkx2.1-cre+/Rosa-DTA* mice (*Figure 5C* and *Figure 5—figure supplement 2C*; n=5). The vessels displayed a significant reduction in the number of nodes and branches and of the density of the vascular network (reduction of nodes $16.23 \pm 4.33\%$, p<0.05; reduction of branches $12.93 \pm 2.37\%$, p<0.05; reduction of the total volume of the vascular network: $21.8 \pm 1.98\%$, p<0.05; n=4, unpaired Student's *t*-test, *Figure 5C,G* and *Table 1*). Moreover, the number of erythrocyte aggregates was significantly increased too (*Figure 5F,H* and *Table 2*; n=5, unpaired Student's *t*-test). These secondary defects may reflect brain vessel dysfunction as a result of disturbed brain vessel network development.

As in *Nkx2.1-cre+/Rosa-DTA* mice, the involvement of pericytes can be directly excluded as a causative factor of the observed vasculature defects in *Cspg4-cre+/Rosa-DTA* due to lack of recombination in this cell population (*Figure 3B and 3F*). The only population that is centrally affected in

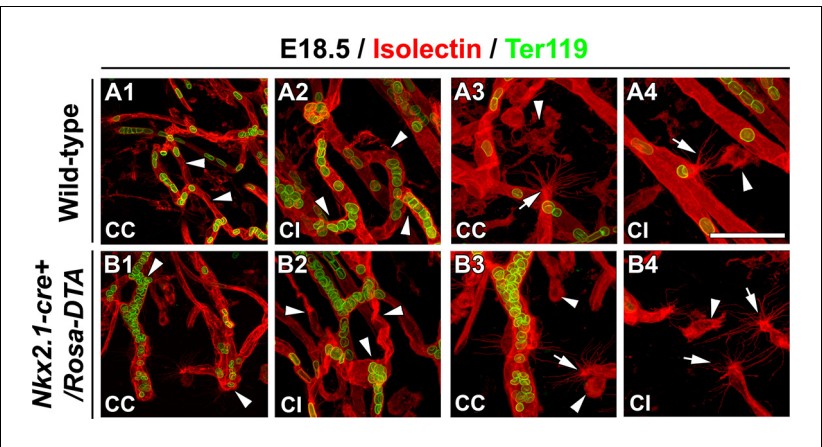

**Figure 6.** Macrophages and tip cells are not affected in *Nkx2*-1Cre[+]/Rosa-DTA mice. (**A–B**) Double immunohistochemistry for Isolectin and Ter119, to visualize erythrocytes, on 250-µm-thick coronal of corpus callosum (CC) and cingulate bundle (CI) sections in wild-type (**A1–A4**) (n=5) and *Nkx2.1-cre[+]/Rosa-DTA* (**B1–B4**) (n=5) mice at E18.5. In the CC and the CI (**B1–B2**) of *Nkx2.1-cre[+]/Rosa-DTA* mice, the blood vessels have a twisted shape and the erythrocytes are clustered (white arrowheads in **B1** and **B2**) compared to the wild-type vessels that formed a regular network (white arrowheads in **A1** and **A2**). In the CC and the CI of both wild-type (**A3–A4**) and *Nkx2.1-cre[+]/Rosa-DTA* mice (**B3–B4**), guidepost macrophages labeled by the isolectin (white arrowheads in **A3–A4** and **B3–B4**) are found in the close vicinity of the tip cells (white arrows in **A3–A4** and **B3–B4**). The tip cells exhibit the same morphology and the same number of filopodia with similar length in both circumstances. Bar = 60 µm in **A1, B1**; 40 µm in **A2, B2, A3, A4, B3, B4**.

both transgenic mice strains, *Cspg4-cre[+]/Rosa-DTA* and *Nkx2.1-cre[+]/Rosa-DTA* mice, are NG2[+] glia. Hence, the poor development of the vascular network and exactly similar vascular defects in both transgenic mice strains, *Cspg4-cre[+]/Rosa-DTA* and *Nkx2.1-cre[+]/Rosa-DTA* mice, is triggered by the absence of NG2[+] glia. Thus, confirming the involvement of *Nkx2.1*-derived NG2[+] glia in controlling brain angiogenesis.

We also noticed some axonal guidance defects in the telencephalic commissures of *Nkx2.1-cre[+]/Rosa-DTA* mice (*Minocha et al., 2015*). As these defects were not at all visible in *Cspg4-cre[+]/Rosa-DTA* mice, the involvement of NG2[+] glia could be excluded.

Analyses of postnatal *Cspg4-cre[+]/Rosa-DTA* brains at P17-P21 (n=3), however, did not reveal any significant blood vessel network defects when compared to littermate *Cspg4-cre[-]/Rosa-DTA* brains (n=3) (*Figure 5—figure supplement 4*). Thus, it appears that the contribution of the NG2[+] glia is important to keep up with the fast pace of embryonic brain development; however, their loss in postnatal brain is may be compensated by either later appearing waves of NG2[+] glia like shown previously (*Kessaris et al., 2006*) or through other cells.

Taken together, this study shows for the first time that NG2[+] glia play a major role in embryonic brain angiogenesis. The temporal and spatial presence of these NG2[+] glia is specifically articulated to aid proper vessel network formation during embryogenesis. The poor development of the blood vessel network in absence of NG2[+] glia underlines the importance of understanding the point-to-point connectivity that is generated by the contact between vessels and polydendrocyte processes.

## Conclusions

Our results, altogether, reveal that brain angiogenesis in embryos requires the presence of *Nkx2.1*-derived NG2[+] glia. The *Nkx2.1*-derived NG2[+] glia begin to populate the entire telencephalon by E14.5 but are transient in nature and disappear by the early postnatal stages. The astrocytes and macrophages have already been reported to play divergent roles during angiogenesis (*Fantin et al., 2010*; *Larrivee et al., 2009*; *He et al., 2013*). This study, however, shows for the first time that *Nkx2.1*-derived NG2[+] glia play a major role in brain angiogenesis. This unravels the important interplay between the ventral and the dorsal telencephalon in multiple physiological functions. We show here that the ventral telencephalon participates not only to the establishment of the cortical

**Table 1.** Nodes, branches and volume of the vascular network visualized per CCi section in control and transgenic mice used to ablate *Nkx2.1*-derived or only NG2[+] cells. The values (mean ± SEM) corresponding to the number of nodes, the number of branches and the volume of the vascular network per CCi section are given for: (1) *Nkx2.1-cre[+]/Rosa-DTA* mice and their corresponding control *Nkx2.1-cre[-]/Rosa-DTA* mice, and (2) *Cspg4-cre[+]/Rosa-DTA* mice and their corresponding control *Cspg4-cre[-]/Rosa-DTA* mice (n=4 each; unpaired Student's *t*-test).

| | Values (mean ± SEM) for the vascular network per CCi section | | |
| --- | --- | --- | --- |
| Genotype | Nodes number | Branches number | Vascular volume * 10$^3$ ($\mu$m$^3$) |
| *Nkx2.1-cre[-]/Rosa-DTA* | 103.44 ± 7.60 | 241.77 ± 14.65 | 20302.06 ± 1290.76 |
| *Nkx2.1-cre[+]/Rosa-DTA* | 84.63 ± 6.65 | 207.75 ± 11.56 | 16692.46 ± 879.03 |
| *Cspg4-cre[-]/Rosa-DTA* | 135.50 ± 8.60 | 299.50 ± 16.89 | 20361.28 ± 1429.69 |
| *Cspg4-cre[+]/Rosa-DTA* | 109.25 ± 6.20 | 250.87 ± 13 | 15921.88 ± 403.44 |

neuronal circuitry by generating tangentially migrating GABAergic interneurons, but also to embryonic angiogenesis. By revealing a novel and essential role for NG2[+] glia originating from the ventral telencephalon in cortical angiogenesis, our study brings new perspectives to pathophysiological consequences linked to tangential migration defects.

## Discussion

### NG2[+] glia participate to proper embryonic angiogenesis

The generation of brain's vascular network starts very early during embryogenesis, and it occurs concomitantly with the development of neuronal network in the brain (*Patan, 2000*; *Vasudevan and Bhide, 2008*). Efficient growth of the vasculature is essential to meet the metabolic needs of the developing brain. However, the mechanisms regulating vessel network's establishment are not yet fully understood. Several studies, so far, have shown that angiogenesis involves participation of ECs along with perivascular cells, referred to as pericytes, and macrophages (*Fantin et al., 2010*; *Newman and Hughes, 2012*; *Nucera et al., 2011*; *Outtz et al., 2011*; *Pollard, 2009*; *Bergers and Song, 2005*). This angiogenic process is completed by a complex repertoire of spatially and temporally orchestrated events that engage several transcription factors, growth factors with their receptors, adhesion molecules, proteases, chemokines, cytokines, and extracellular matrix (*Adams and Alitalo, 2007*; *Carmeliet and Tessier-Lavigne, 2005*; *Lu et al., 2004*; *Larrivee et al., 2007*; *Eichmann et al., 2005*).

In our model to study embryonic angiogenesis, we have used cell-specific ablation strategy with the help of Cre reporter strains, namely *Nkx2.1-cre* and *Cspg4-cre* mice, together with 'floxed'

**Table 2.** Number of erythrocyte clusters and of individual erythrocytes visualized per CCi section in control and transgenic mice used to ablate *Nkx2.1*-derived or only NG2[+] cells. The values (mean ± SEM) corresponding to the number of erythrocyte clusters and the number of individual erythrocytes per CCi are given for: (1) *Nkx2.1-cre[+]/Rosa-DTA* mice and their corresponding control *Nkx2.1-cre[-]/Rosa-DTA* mice, and (2) *Cspg4-cre[+]/Rosa-DTA* mice and their corresponding control *Cspg4-cre[-]/Rosa-DTA* mice (n=5 each; unpaired Student's *t*-test).

| | Values (mean ± SEM) per CCi section | |
| --- | --- | --- |
| Genotype | Number of erythrocyte clusters per CCi section | Number of individual erythrocytes per CCi section |
| *Nkx2.1-cre[-]/Rosa-DTA* | 6.60 ± 0.88 | 302.33 ± 19.12 |
| *Nkx2.1-cre[+]/RosaDTA* | 13.00 ± 2.62 | 273.42 ± 41.44 |
| *Cspg4-cre[-]/Rosa-DTA* | 5.83 ± 0.62 | 397.80 ± 18.96 |
| *Cspg4-cre[+]/Rosa-DTA* | 14.95 ± 1.77 | 447.30 ± 22.91 |

diphtheria toxin (Rosa-DTA) to selectively kill cells under Nkx2.1 or NG2 promoter, respectively. With *Nkx2.1-cre⁺/Rosa-DTA* mice, we observed the formation of a poorly developed vessel network in several telencephalic regions. A significant reduction of intersections, connectivity and vasculature density was observed. The cortical vessels were seen to loose their regular diameter and harbored erythrocyte accumulations at the level of enlarged vessel segments. Since, the *Nkx2.1*-derived cells comprise of GABAergic interneurons (*Marin and Rubenstein, 2001*; *Xu et al., 2008*; *Sussel et al., 1999*; *Du et al., 2008*; *Xu et al., 2004*) and NG2⁺ glia, we investigated the two populations separately to dissect the responsible cell type. With *Nkx2.1-cre⁺/Rosa-DTA* mice, a drastic decrease of 50% GABAergic interneurons and complete loss of NG2⁺ glia was seen at midline regions. Complete loss of NG2⁺ glia with *Nkx2.1-cre⁺/Rosa-DTA* mice displayed the subpallial origin of the polydendrocyte population at the midline specifically.

Further on, to look into the role of GABAergic interneurons in the observed vessel network phenotype, we made use of *Nkx2.1-cre⁺/Eno2-DTA:Gad1-EGFP* mice. These mice did not show any loss of NG2⁺ glia nor any vessel network formation defects, although the loss of GABAergic neurons was comparable to that seen with *Nkx2.1-cre⁺/Rosa-DTA* mice. Thus, we could exclude the GABAergic interneurons from being the cause of the observed phenotype.

We could also exclude the role of astrocytes because *Nkx2.1-cre⁺*mice recombination can only been seen in GLAST⁺ astrocytes of the ventral telencephalon, and dorsal telencephalon astrocytes remain unaffected in *Nkx2.1-cre⁺/Rosa-DTA* mice.

Next, we aimed to target the NG2⁺ population. To this purpose, we additionally used *Cspg4-cre⁺/Rosa-DTA* mice where NG2⁺ glia were depleted by more than 50%. Interestingly, we visualized similar vessel network defects as *Nkx2.1-cre⁺/Rosa-DTA* mice. Notably, NG2 chondroitin sulfate proteoglycan is expressed not only in glia but also on surface of pericytes in the brain (*Nishiyama et al., 2009*; *Ozerdem et al., 2001*). The loss of pericytes can also lead to aberrations in fetal brain vasculature and maintenance of blood–brain barrier leading to vascular instability (*Stallcup and Huang, 2008*; *Armulik et al., 2010*; *Daneman et al., 2010*). Since the role of both perivascular cells has been well established in angiogenesis, we investigated if our *Nkx2.1-cre⁺/Rosa-DTA* and *Cspg4-cre⁺/Rosa-DTA* mice lack pericytes or ECs. The *Nkx2.1-cre⁺/Rosa-DTA* mice directly excluded the involvement of pericytes and ECs due to the absence of Cre expression in both cell types. As a result, we can exclude that the defects can be due to the intrinsic action of Nkx2.1 in ECs as observed in Nkx2.1 KO mice (*Vasudevan and Bhide, 2008*; *Vasudevan, et al., 2008*). Also, the *Cspg4-cre⁺/Rosa-DTA* mice did not show any loss of pericytes due to very low levels for Cre recombination in these cells. Therefore, our cell-specific ablation analysis directly points toward involvement of NG2⁺ glia, the only common affected cell population, in embryonic brain angiogenesis. The origin of these NG2⁺ glia coincides not only temporally with the brain vessel network formation but also spatially due to their close proximity to the developing vessel network.

Another identified key player in brain vessel network are macrophages that have been implicated in both physiological and pathological angiogenesis (*Newman and Hughes, 2012*; *Nucera et al., 2011*; *Pollard, 2009*). The macrophages colonize the embryonic brain even before vascularization is built (*Fantin et al., 2010*) and have been observed to localize at vessels junctions and to interact with the endothelial tip cells, by forming sort of bridges in order to align them and to prepare them for the fusion (*Fantin et al., 2010*). Similar to macrophages, the NG2⁺ glia were localized at sprouting tip cells or at branch fusion points, but, in addition, were also present all along the vessel walls. They made several contacts with mural cells with their extended processes. Although our selective cell ablation strategy did not affect the macrophage population, it is probable that the two populations may influence the vessel network through a concerted action.

## Potential molecular mechanisms used by NG2⁺ glia to control brain angiogenesis

The strategic positioning of NG2⁺ glia raises the possibility that they interact functionally by making synapses and generating action potentials (*De Biase et al., 2010*; *Clarke et al., 2012*) or by sending signals to mural cells. Their positioning along and around the developing networks might provide them with the proximity required to monitor the firing patterns of surrounding cells (*De Biase et al., 2010*; *Clarke et al., 2012*).

Additionally, the NG2⁺ glia may act as guidepost cells by secreting growing factors or guidance molecules required for the proper development and function of brain vessels. During

development, the formation of the vessel network is dependent on several growth factors like vascular endothelial growth factor A (VEGF-A), VEGF-B, VEGF-C, VEGF-D, and placental growth factor (PlGF) that bind to different tyrosine kinase receptors (VEGFR) expressed on ECs (*Adams and Alitalo, 2007*; *Coultas et al., 2005*; *Fantin et al., 2010*; *Ferrara et al., 2003*; *Gerhardt et al., 2003*; *Grunewald et al., 2006*; *Laakkonen et al., 2007*; *Lee et al., 2005*; *Neufeld et al., 2002*; *Pan et al., 2007*; *Plate, 1999*; *Ruhrberg et al., 2002*; *Shibuya, 2009*). In addition to these growth factors, other signaling pathways also aid in the development of the vascular circuitry like angiopoietin1/Tie2; Notch/Delta-like-4/JAG1; Wnt signaling and guidance cues with their associated receptors such as ephrin-B2 and EphB4, Slit and Robo4, Netrins and Unc5/Dcc, class 3 semaphorins and Neuropilins/Plexins (*Adams and Alitalo, 2007*; *Adams and Eichmann, 2010*; *Bedell et al., 2005*; *Bray, 2006*; *Eichmann et al., 2005*; *Gerhardt et al., 2004*; *Gitler et al., 2004*; *Gu et al., 2005*; *Hellstrom et al., 2007*; *Klagsbrun et al., 2002*; *Kruger et al., 2005*; *Larrivee et al., 2009*; *Leslie et al., 2007*; *Lobov et al., 2007*; *Lu et al., 2004*; *Neufeld et al., 2005*; *Park et al., 2003*; *Sainson et al., 2005*; *Suchting et al., 2007*; *Torres-Vazquez et al., 2004*; *Wilson et al., 2006*).

Recently, an elegant study performed in postnatal mice showed that the OPCs-intrinsic Hypoxia-inducible factors 1/2 (HIF1/2) signaling stimulates EC proliferation in vitro and increases CC white matter angiogenesis in vivo (*Yuen et al., 2014*). Conversely, OPCs-encoded HIF1/2a inactivation causes insufficient CC angiogenesis and commissural axon's degeneration. This study leads us to question if the NG2$^+$ embryonic glia we have identified also regulate vessels outgrowth through HIF1a and/or HIF2a action. Notably, the population addressed by *Yuen et al., 2014* primarily focuses on the effects in postnatal brains, whereas the NG2$^+$ glia identified by us disappear by P8. Nonetheless, it will be interesting to perform in vivo brain analyses of angiogenesis in mouse embryos from *Cspg4-cre* mice crossed with *HIF1a floxed*, *HIF2a floxed* or *HIF1/2a floxed* mice, allowing selective gene inactivation of HIFs in NG2$^+$ glia only.

Several studies have shown that growth factors can regulate the NG2$^+$ glial cell proliferation and migration – of which some belong to PI3K/mTOR and Wnt/β-catenin signaling pathways (*Hill et al., 2013*; *Frost et al., 2009*). In particular, Wnt/β-catenin signaling pathway has been suggested to be the important player required for molecular and cellular steps necessary for proper brain vascularization. The broad expression of Wnt7a/7b ligands in the developing early embryonic CNS and induction of Wnt/ β-catenin signaling is required for formation and differentiation of the embryonic brain vasculature (*Stenman et al., 2008*). Additionally, the Wnt/ β-catenin signaling is also important for the blood–brain barrier, a structure formed by the brain blood vessels. The effects of the absence of Wnt7a/7b ligands are evident already in early embryonic ages, that is E12.5, and are crucial for ventral CNS vasculature. Notably, the NG2$^+$glia are not present in the dorsal telencephalon as early as E12.5. Hence, a more closer dissection of these and other Wnts would be required to ascertain if the NG2$^+$glia are mediating their roles in vascularization through Wnts at later stages, that is, when we observed the defects (E16.5–E18.5).

Since the repertoire of factors expressed by NG2$^+$ glia has not been explored much yet, a complete dissection of the different angiogenic factors will be required for a better understanding. It would be also necessary to analyze the coordinated and oriented cell movements that are necessary for the development of the vascular network especially during sprouting and subsequent maturation phases. The complete understanding of molecular mechanisms undertaken by NG2$^+$ glia to guide and aid angiogenesis will provide significant insight into angiogenesis.

## Spatial organization of NG2$^+$ glia

This study highlights the importance of spatially and temporally synchronized generation of NG2$^+$ glia that act as guidepost cells for embryonic vasculature establishment. The migratory profile of the NG2$^+$ glia has been explored in several contexts before, but there is no such available information if they directly use the vasculature. Like the other major population of *Nkx2.1*-derived GABAergic interneurons (*Marin and Rubenstein, 2001*; *Guo and Anton, 2014*; *Elias et al., 2008*; *Chedotal and Rijli, 2009*), we believe that the NG2$^+$ glia also use tangential migration to colonize the cortex. Clues for this are evident from our immunostaining of the NG2$^+$ glia at embryonic and early postnatal stages in the control brains. Previous reports have discussed the migration of sub-populations of the NG2$^+$ cells during embryonic and early postnatal development (*Sugimoto et al., 2001*; *Cayre et al., 2009*; *Aguirre and Gallo, 2004*). Additionally, in adult tissues, the migration of

NG2[+] cells has also been observed where these cells migrate in relation to each other (*Hughes et al., 2013*).

Also, some cell–cell interactions between axonal tracts and OPCs has been believed to be important for migration, and molecules such as N-cadherins, neuregulins, ephrins, and integrins have been implicated (*de Castro and Bribian, 2005*). A recent study showed that the proteoglycan NG2 itself could play a role in the directed localization of NG2[+] cells (*Biname et al., 2013*). Fittingly, the NG2[+] cells exhibit maximal migration upon encountering a region deficient of other NG2[+] cells (*Cayre et al., 2009*) or when the region has been induced to lack NG2[+] cells either through experimental intervention or as a result of injury (*Blakemore and Irvine, 2008*).

## NG2[+] glia and possible involvement in pathophysiological mechanisms

Our work depicting the elucidation of the role of NG2[+] glia in early phases of brain vessel network provides new insights into the mechanisms that take place during embryonic brain angiogenesis. The requirement of a properly developed vessel network is immense during embryonic ages to meet the growing nutritional needs of the embryo. Several human neuropathologies related to neonatal encephalopathy are induced by incidents involving hypoxia to the brain (e.g. in cerebral palsy). We suspect that one possible reason for cerebral palsy could be the presence of dysfunctional NG2[+] glia, which then leads to decrease in vascularization followed by neuronal injury generating the pathological condition.

OPCs have been shown to be involved in glioblastoma genesis and evolution (*Barrett et al., 2012*; *Lindberg et al., 2009*; *Persson et al., 2010*; *Sugiarto et al., 2011*; *Svendsen et al., 2011*). Therefore, our research may help in the understanding of the pathological cellular and vessel processes that take place during brain tumor evolution. Strategies based on NG2[+] cell targeting may significantly improve long-term outcome of patients with glioblastoma (GBM). It will be interesting to further characterize the role of NG2[+] glia in GBMs tumour angiogenesis and to investigate whether NG2[+] glia-encoded HIFs are involved in tumour-associated NG2[+] glial differentiation and angiogenesis.

Altogether, the combined role play of ECs, pericytes, macrophages, and glia and the molecular mechanisms defining their respective mode of actions can aid in better understanding of the blood vessel network formation, and help alleviate many disease pathologies.

## Materials and methods

### Animals

All studies on mice of either sex have been performed in compliance with the national and international guidelines and with the approval of the Federation of Swiss cantonal Veterinary Officers (2164). For staging of embryos, midday of the day of vaginal plug formation was considered as embryonic day 0.5 (E0.5). WT mice maintained on a *C57Bl/6* genetic background were used for developmental analysis. We used heterozygous *Gad1-EGFP* knock-in mice, described in this work as *Gad1-EGFP* mice (*Tamamaki et al., 2003*). We used *Nkx2.1-cre* (*Xu et al., 2008*) and *Cspg4-cre* (Jackson Laboratory: *B6;FVB-Tg(Cspg4-cre)1Akik/J*) (*Zhu et al., 2008a*) transgenic mice that have been described previously. The reporter mouse *Rosa26R–Yellow fluorescent protein (YFP)* (*Srinivas et al., 2001*) was used to reliably express YFP under the control of the Rosa26 promoter upon Cre-mediated recombination. Embryos were recognized by their YFP fluorescence. The reporter *Rosa26:lacZ/DTA (Rosa-DTA)* mouse line (*Brockschnieder et al., 2006*) was used to express conditionally the cytototoxic diphtheria toxin polypeptide toxic fragment A (DTA) allele under the control of ubiquitously active Rosa26 promoter. By crossing *Nkx2.1-cre* mice with *Rosa26-DTA* mice that express the diphtheria toxin under the control of the Nkx2.1 promoter, only Nkx2.1-expressing post-mitotic cells were depleted in the whole brain without affecting the Nkx2.1[+] precursors in the progenitor zones. The neuron-specific enolase (*Eno2*)-stop-DTA (*Eno2-DTA*) mice (*Kobayakawa, et al., 2007*) were used to induce the expression of highly potent diphtheria toxin fragment A (DTA) from neuron-specific enolase locus, and resulted in specific ablation of neurons only.

## Immunohistochemistry

Embryos were collected after caesarean section and quickly killed by decapitation. Their brains were dissected out and fixed by immersion overnight in 4% paraformaldehyde solution in 0.1 M phosphate buffer (pH 7.4) at 4°C. Postnatal mice were profoundly anesthetized and perfused with the same fixative, and their brains post-fixed for 4 hr. Brains were cryoprotected in a solution of 30% sucrose in 0.1 M phosphate buffer (pH 7.4), frozen and cut in 50-μm-thick coronal sections for immunostaining.

Rat monoclonal antibodies used were: Lymphocyte antigen76 (Ter119) (Lifespan Bioscience, Switzerland); PDGFRα (BD Bioscience, Switzerland); PECAM (CD31) (BD Pharmingen, Switzerland). Rabbit polyclonal antibodies used were: GFAP (DAKO, Carpinteria, CA); GFP (Molecular Probes, Zug, Switzerland); NG2 (Chemicon, Temecula, CA); Nkx2.1 (Biopat, Caserta, Italy); Olig2 (Millipore, Switzerland); S100β (Swant, Bellinzona, Switzerland). Goat polyclonal antibodies used were: PDGFRβ (R&D Systems, Switzerland). Guinea pig antibody was: GLAST (Chemicon, Temecula, CA). Chicken antibody was: GFP (Aves, UK). Biotin-conjugated antibodies were: Isolectin Ig-IB4 (Molecular probes, Zug, Switzerland).

Fluorescence immunostaining: Non-specific bindings were blocked during pre-incubation and incubations after adding 2% normal horse serum in PBS 1X solution supplemented with 0.3% Triton X-100. The primary antibodies were detected with Cy3-conjugated (Jackson ImmunoResearch laboratories, West Grove, PA) and Alexa 488-, Alexa 594- or Alexa 647-conjugated antibodies or streptavidin (Molecular Probes, Eugene, OR). Sections were counterstained with Hoechst 33,258 (Molecular Probes, Switzerland), mounted on glassslides and covered in Mowiol 4–88 mounting medium (Calbiochem, Bad Soden, Germany).

DAB immunostaining: Endogenous peroxidase reaction was quenched with 0.5% hydrogen peroxide in methanol, and unspecific binding was blocked by adding 2% normal horse serum during incubations in the Tris-buffered solutions containing 0.3% Triton X-100. The primary antibodies were detected with biotinylated secondary antibodies (Jackson ImmunoResearch, West Grove, PA) and the Vector-Elite ABC kit (Vector Laboratories, Burlingame, CA). The slices were mounted on glassslides, dried, dehydrated, and covered with Eukitt mounting medium.

DAB immunostaining was adapted for erythrocytes staining: Endogenous peroxidase reaction was directly used to visualize the erythrocytes. Therefore, no hydrogen peroxide was added. Sections were pre-incubated for 10 min in 0.05% DAB diluted in 0.1 M Tris buffer. The reaction was started with the same solution supplemented with 0.5% $H_2O_2$ and stopped with 0.1 M Tris buffer after 10 min. The slices were mounted as per standard protocol.

## Imaging

DAB-stained sections were imaged with a Zeiss Axioplan2 microscope equipped with 10×, 20×, 40×, or 100× Plan neofluar objectives and coupled to a CCD camera (Axiocam MRc 1388x1040 pixels). Fluorescent-immunostained sections were imaged using confocal microscopes (Leica SP5 or Zeiss LSM 710 Quasar) equipped with 10×, 20×, 40× oil Plan neofluar and 63× oil Plan apochromat objectives. Fluorophore excitation and scanning were done with an Argon laser 458, 488, 514 nm (blue excitation for GFP and Alexa 488), with a HeNe laser 543 nm (green excitation for Alexa 594, CY3 and DiI), with a HeNe laser 633 nm (excitation for Alexa 647 and CY5), and with a Diode laser 405 nm (for Hoechst-stained sections). Z-stacks of 10–25 planes were acquired for each coronal section in a multitrack mode avoiding crosstalk, and the creation of isosurfaces was done with Imaris 7.2.1 software (Bitplane Inc.). All 3D Z-stack reconstructions and image processing's were performed with Imaris 7.2.1 software. The generation of iso-surfaces (object defining a surface surrounding voxels located between two threshold values) allowed us to visualize the contours of cells, of blood vessels and of growth cones. The colocalization between two fluorochromes was calculated and visualized by creating a yellow channel. Figures were processed in Adobe Photoshop CS5.

## Quantifications

*Loss of GABAergic neurons*: In 50-μm-thick coronal sections of *Nkx2.1-cre$^+$/Rosa-DTA:GAD67-GFP, Nkx2.1-cre$^-$/Rosa-DTA:GAD67-GFP, Nkx2.1-cre$^+$/Eno2-DTA:Gad1-EGFP* and *Nkx2.1-cre$^-$/Eno2-DTA: Gad1-EGFP* embryos, GABAergic neurons of the cortical region (CCi and CI) were counted at E18.5 as the number of cells labeled for the GFP. Quantification were done on a sample of n=8 sections in

*Nkx2.1-cre[+]/Rosa-DTA:Gad1-EGFP* and corresponding controls and on a sample of n=6 sections in *Nkx2.1-cre[+]/Eno2-DTA:Gad1-EGFP* and corresponding controls. To study the density of GABAergic neurons, the values were quantified in all the scanned planes of each stack and were reported per volume unit (number of cells/mm$^3$). The quantifications were done using the Imaris 7.2.1 software.

*Preservation of GLAST[+] glia in the cingulate cortex*: In 50-μm-thick brain sections, the GLAST$^+$ glia in the CCi labeled for GLAST were counted from a sample of n=4 sections of E18.5 *Nkx2.1-cre[-]/Rosa-DTA* and of *Nkx2.1-cre[+]/Rosa-DTA* embryos.

To study the density of GABAergic neurons and GLAST$^+$ glia, the values were quantified in all the scanned planes of each stack and were reported per volume unit (number of cells/mm$^3$). The quantifications were done using the Imaris 7.3.1 software.

*Cspg4-cre recombination level analysis and percentage of NG2, Olig2, S100β, GFAP and GLAST markers expression by YFP-labeled NG2[+] glia in Cspg4-cre[+]/Rosa-EYFP mice*: In 50-μm-thick brain sections of *Cspg4-cre[+]/Rosa-EYFP* embryos at E18.5, glia of the CC (midline), CI and CCi labeled for YFP and NG2 were counted (n=3). For each condition, at least 3 different Z-stacks were obtained at 63× magnification by using a Leica SP5 microscope. To study the density of NG2$^+$ glia, the values were quantified in all the scanned planes of each stack and were reported per volume unit (number of cells/mm$^3$).

*Cspg4-cre recombination level analysis and percentage of PDGFRβ pericyte markers expression by YFP-labeled NG2[+] cells in Cspg4-cre[+]/Rosa-EYFP mice*: In 50-μm-thick brain sections of *Cspg4-cre[+]/Rosa-EYFP* embryos at E18.5 (n=10), YFP+ pericytes and YFP+ pericytes of the dorsal telencephalic regions labeled for NG2 were counted.

The percentage of Cre-mediated recombination within the NG2$^+$ cells was calculated as follows: YFP$^+$ cells/NG2$^+$ cells X100. The percentage of YFP$^+$/PDGFRβ$^+$, YFP$^+$/NG2$^+$, Olig2$^+$/NG2$^+$, YFP$^+$/S100β$^+$, YFP$^+$/GFAP$^+$ and YFP$^+$/GLAST$^+$ glia within YFP$^+$ cells was calculated. The quantification was done using Imaris 7.2.1 software.

*Percentage of NG2 marker expression by YFP-labeled cells in Nkx2.1-cre[+]/Rosa-EYFP mice*: In 50-μm-thick brain sections of *Nkx2.1-cre[+]/Rosa-EYFP* embryos at E18.5, YFP$^+$ cells of the CI labeled for the NG2 were counted (n=3). For each condition, at least 3 different Z-stacks were obtained at 63x magnification by using a Leica SP5 microscope. To study the density of YFP$^+$/NG2$^+$ glia, the values were quantified in all the scanned planes of each stack and were reported per volume unit (number of cells/mm$^3$). The percentage of YFP$^+$/NG2$^+$ glia within YFP$^+$ cells was calculated. The quantification was done using Imaris 7.2.1 software.

*Loss of NG2[+] glia and PDGFRβ[+] pericytes*:In 50-μm-thick brain sections, the NG2$^+$ glia in the CI labeled for NG2 were counted from a sample of n=11 sections of E18.5 *Cspg4-cre[-]/Rosa-DTA* and of *Cspg4-cre[+]/Rosa-DTA* embryos. Similarly, the pericytes in the dorsal telencephalon labeled for PDGFRβ were counted from a sample of n=11 sections of *Cspg4-cre[-]/Rosa-DTA* and of *Cspg4-cre[+]/Rosa-DTA* mice. To study the density of NG2$^+$ glia, PDGFRβ$^+$ pericytes in each condition, the values were quantified in all the scanned planes of each stack and were reported per volume unit (number of cells/mm$^3$). The quantification was done using Imaris 7.2.1 software.

*Blood vessels morphology and erythrocytes distribution analysis*:In 50-μm-thick brain sections of *Nkx2.1-cre[+]/Rosa-DTA*, *Nkx2.1-cre[-]/Rosa-DTA*, *Cspg4-cre[+]/Rosa-DTA* and *Cspg4-cre[-]/Rosa-DTA* embryos at E18.5, the vessels network morphology and erythrocytes were analyzed from a sample of more that n=10 sections for each condition.

DAB staining for PECAM was used to label blood vessels. The blood vessels morphology was analyzed in a predetermined region comprising of the CC, CI, and CCi (surface area/section=6*10$^6$μm$^2$). The 3D blood vessels morphology was reconstructed using Neurolucida 9.0 software (MicroBrightField, Williston, VT). The axonal tracing tool allows to trace and to calculate for each vessels network the number of individual trees, the number of nodes, the number of branches, the length, the surface, and the volume of the blood vessels network for each condition.

DAB-stained erythrocytes were counted in a predetermined region comprising of the CC, CI, and CCi (surface area/section=20*10$^6$μm$^2$). The erythrocytes were counted as a cluster when more than five consecutive erythrocytes were closely apposed to each other, while the well separated individual erythrocytes were counted as individual cells. The quantification was done using Neurolucida 9.0 software.

## Statistical analysis

The results from all quantifications were analyzed with the aid of Statview software (SAS Institute). For all analysis, values from at least three independent experiments were first tested for normality. Values that followed a normal distribution were compared using Student's *t*-test (*p<0.05, **p<0.01, ***p<0.001).

## Atlas and nomenclature

The neuroanatomical nomenclature is based on the 'Atlas of the prenatal mouse brain' (*Schambra et al., 1991*).

## Acknowledgements

We are particularly grateful to C. Devenoges for technical assistance. We would like to thank F. Thevenaz and A. Gnecchi for mouse care, plugs and genotyping. We thank J-Y. Chatton from the Cellular Imaging Facility (CIF, University of Lausanne, Switzerland) for imaging assistance. We thank Pr D. Riethmacher for the gift of the *Rosa-DTA* mice (Zentrum für Molekulare Neurobiologie, University of Hamburg, Germany) and S. Itohara (RIKEN Brain Science Instituta, Japan) for the gift of the *Eno2-DTA* mice. S. Minocha was supported by a postdoctoral fellowship of the Fondation Pierre Mercier pour la science. The work done in the laboratories of C. Lebrand and J-P. Hornung was supported by funds from Swiss National Foundation Grant # 31003A-122550.

## Additional information

### Funding

| Funder | Grant reference number | Author |
| --- | --- | --- |
| Swiss National Foundation | Grant 31003A-122550 | Cecile Lebrand |
| Fondaton Pierre Mercier | | Shilpi Minocha |

The funders had no role in study design, data collection and interpretation, or the decision to submit the work for publication.

### Author contributions

SM, Conception and design, Acquisition of data, Analysis and interpretation of data, Drafting or revising the article; DV, Conception and design, Acquisition of data, Analysis and interpretation of data; IB, AE, J-PH, Conception and design, Drafting or revising the article; CL, Conception and design, Analysis and interpretation of data, Drafting or revising the article

### Author ORCIDs

Jean-Pierre Hornung, http://orcid.org/0000-0002-9229-7520
Cecile Lebrand, http://orcid.org/0000-0002-2750-3164

### Ethics

Animal experimentation: All studies on mice of either sex have been performed in compliance with the national and international guidelines and with the approval of the Federation of Swiss cantonal Veterinary Officers (2164).

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
