## [Decision Letter]

Thank you for submitting your work entitled "NG2 glial cells are required for vessel network formation during embryonic development" for peer review at *eLife*. Your submission has been favorably evaluated by K VijayRaghavan (Senior editor), Joseph G. Gleeson (Reviewing editor), and two reviewers, one of whom,

Akiko Nishiyama, has agreed to reveal her identity.

The reviewers have discussed the reviews with one another and the Reviewing editor has drafted this decision to help you prepare a revised submission.

Overall, the manuscript was favorably reviewed. However, a number of points were raised that need to be addressed (by reformatting and in the Discussion) before final acceptance can be offered.

The Reviewing editor and the other reviewers discussed their comments before we reached this decision, and the Reviewing editor has assembled the following comments to help you prepare a revised submission.

Summary:

In this manuscript, Minocha et al. have investigated a role for NG2 glial cells in orchestration of embryonic forebrain vasculature. The authors demonstrate that NG2^+^ glia are closely associated with developing vascular network in E12.5-E18.5 dorsal telencephalon, using immunohistochemistry and reporter expression from *Nkx2.1-cre:ROSA-YFP* and *NG2-cre:ROSA-YFP* transgenic mice. In this manuscript this part of the data is strongly supported by high-quality confocal and DAB images. The data shows intriguing associations between migrating NG2 cells and the developing vasculature, and that death of ventral NXK2.1 progenitor cells, which include migrating ventral NG2 cells with their destination in cortex, results in clear abnormalities of cortical vasculature. This is done using *Nkx2.1-cre:ROSA-floxed-DTA* and *NG2-cre:ROSA-floxed-DTA* mice. They show that the number of nodes and branches and the density of the vascular network are all decreased in these mice that lack polydendrocytes. From these findings, they conclude that polydendrocytes are critical for angiogenesis in the dorsal telencephalon. The major strength is showing that NG2 cells (or oligodendrocyte precursors (OPCs)) have a role in embryonic cortical vasculature, but as a similar role has been described for OPCs of early postnatal cortex (Yuen et al., 2014),

Much has been learned in the past decade about the development and fate of polydendrocytes but their developmental and physiological role besides oligodendrocyte production has remained unclear. In this regard, this article asks an important question of whether polydendrocytes play a part in angiogenesis, and this hypothesis is very reasonable based on their observations. The data are clearly presented and are supported by high quality images and some quantification. The authors have convincingly shown that the vascular defects in *Nkx2.1-cre:Rosa-DTA* mice are not caused by loss of interneurons. The first part of the study is very strong. However, there are some questions that should be addressed regarding the cell ablation experiment. Furthermore, there are some confusing aspects of terminologies used, and more importantly, some data is redundant with prior published studies, which needs to be better cited and considered within the novelty of these findings.

Essential revisions:

1) The reviewers were not able to unequivocally conclude that the observed vascular defects are indeed caused by the loss of polydendrocytes. The authors make a convincing case that Cre-mediated recombination does not occur in vascular cells in *NG2-cre:ROSA-YFP* mice. However, other studies have shown Cre activation in embryonic vascular cells in NG2cre mice. Zhu et al., 2008 showed that in NG2-cre:zeg double transgenic mice there is significant reporter expression in vascular cells at E18. Huang et al., GLIA 2014 showed that in *NG2creER:Rosa-tdTomato* mice, which is more sensitive to Cre than ROSA-YFP mice, vascular cells become Tomato+ after Cre induction in the embryo. The lack of YFP detection in pericytes in this study may be due inefficient Cre recombination for activating YFP expression. However, the low level of recombination may still cause sufficient DTA expression to kill pericytes. To rule this out, the authors should perform pericyte staining, e.g. Pdgfrb, in the *NG2-cre:Rosa-DTA* mice to see whether the number and morphology of pericytes are altered after polydendrocytes ablation. The phenotype of *Nkx2.1-cre:Rosa-DTA* still supports the notion that NG2 cells are needed for vascular development. To be complete, the authors should show GLAST staining in these (*Nkx2.1-cre:Rosa-DTA*) mice to rule out the possibility that loss of astrocytes is not contributing to the vascular phenotype. If there is loss of pericytes in *NG2-cre:Rosa-DTA* mice and if there is astrocyte loss in the relevant regions (e.g. corticoseptal boundary as their Minocha et al. 2015 paper suggests), then enthusiasm for the findings will be significantly diminished.

2) No mechanistic detail is provided for how NG2 cells carry out their putative vascular regulatory role. It is unclear whether the authors propose that NG2 cells secrete angiogenic factors and how these balance against known roles for neuroepithelial precursors in embryonic brain (e.g., Stenman et al., 2008,)? For example, it should be straightforward to assess VEGF or Wnt expression at the appropriate stages of NG2 cell development and then target a loss of function study to show the precise mechanisms involved. Do NG2 cells have direct or indirect effects on vasculature? No angiogenic assays are performed.

3) The authors show only one time point for their vascular measurements. Is E18.5 the earliest time point at which they see a difference in the vasculature? Was there a decrease in polydendrocyte numbers prior to that? Do the vascular defects eventually correct themselves after prolonged absence of NG2^+^ glia? A finer parallel time course analysis of the loss of polydendrocytes and the changes in the vasculature should be performed to confirm that the latter follow the former and are not due to non-polydendrocyte-mediated effects of the ablation.

4) The introduction of NG2 cells as "polydendrocytes" is confusing for the general reader. Are the authors referring to OPCs or some other type of NG2 glia? What distinguishes a polydendrocyte from an OPC since both do not only make oligodendrocytes? Until the Results section, the description of NG2-labeled cells fails to mention pericytes. Thus, the description of NG2 cells is somewhat confusing and it becomes important in the paper.

5) The authors state they identify the origins of NG2 glia in the ventral telencephalon. However, the Richardson lab has described this for OPCs in prior studies, importantly Kasaris et al. 2006 Nature Neuroscience, which is not cited. Because NG2 is expressed on OPCs and given findings referred to above, data in Figure 2 seem redundant with prior published work. Indeed, the Kessaris paper showed that early wave OPCs from ventral Nkx2.1^+^ telecephalic precursors are replaced by later waves of OPCs from more dorsal domains. These points and the novelty of the findings need to be clarified.

---

## [Author Response]

Essential revisions:

1) The reviewers were not able to unequivocally conclude that the observed vascular defects are indeed caused by the loss of polydendrocytes. The authors make a convincing case that Cre-mediated recombination does not occur in vascular cells in NG2-cre:ROSA-YFP mice. However, other studies have shown Cre activation in embryonic vascular cells in NG2cre mice. Zhu et al., 2008 showed that in NG2-cre:zeg double transgenic mice there is significant reporter expression in vascular cells at E18. Huang et al., GLIA 2014 showed that in NG2creER:Rosa-tdTomato mice, which is more sensitive to Cre than ROSA-YFP mice, vascular cells become Tomato+ after Cre induction in the embryo. The lack of YFP detection in pericytes in this study may be due inefficient Cre recombination for activating YFP expression. However, the low level of recombination may still cause sufficient DTA expression to kill pericytes. To rule this out, the authors should perform pericyte staining, e.g. Pdgfrb, in the NG2-cre:Rosa-DTA mice to see whether the number and morphology of pericytes are altered after polydendrocytes ablation. The phenotype of Nkx2.1-cre:Rosa-DTA still supports the notion that NG2 cells are needed for vascular development. To be complete, the authors should show GLAST staining in these (Nkx2.1-cre:Rosa-DTA) mice to rule out the possibility that loss of astrocytes is not contributing to the vascular phenotype. If there is loss of pericytes in NG2-cre:Rosa-DTA mice and if there is astrocyte loss in the relevant regions (e.g. corticoseptal boundary as their Minocha et al. 2015 paper suggests), then enthusiasm for the findings will be significantly diminished.

We are aware of the fact that the pericytes are known to physically interact with endothelial cells and contribute towards vessel maintenance and formation. Hence, we wanted to make sure that these cells are not affected in our conditional mutants. Though NG2 is expressed by pericytes, the *NG2-Cre^+^* transgenic strain used by us did not exhibit sufficient Cre-mediated recombination in them. Only very few pericytes, co-labeled for PDGFRβ and YFP, could be seen in the dorsal telencephalon with *NG2-Cre^+^/Rosa-YFP* mice. We had shown some of these results in the previous Figure 3 (additional results now in Figure 3) of the submitted manuscript. Now, we are providing quantification for comparison of pericytes labeled with PDGFRβ and those labeled for YFP in *NG2-Cre^+^/Rosa-YFP* mice. Additionally, the pericytes were not ablated in *NG2-Cre^+^/Rosa-DTA* mice also due to the lack of sufficient Cre-mediated recombination in these cells. We had shown some of these results in the previous Figure 3 (additional results now in Figure 3) of the submitted manuscript.

In light of the concerns raised by the reviewers, we further investigated the *NG2-Cre^+^* strains to confirm the level of recombination efficiency in pericytes by using the following experimental strategies:

A) We took additional images of *NG2-Cre^+^/Rosa-YFP* mice brains stained with PDGFRβ and YFP at E18.5. In all analyzed dorsal telencephalic sections, we could only see very few pericytes that were both PDGFRβ^+^ and YFP^+^. The majority of the PDGFRβ^+^ pericytes population was YFP^-^. Quantifications of the two populations: PDGFRβ^+^ YFP^-^ pericytes and PDGFRβ^+^ YFP^+^ pericytes showed that only 4.95 ± 1.54% of total PDGFRβ^+^ pericytes-population is co-labeled with YFP. Please see quantifications in Figure 3 of the revised manuscript. Thus, these results confirm our previously stated claims that there is scarce Cre-mediated recombination in the pericytes in the *NG2-Cre^+^/Rosa-YFP* mice brains and the large majority of the YFP signal is present in NG2^+^ glia.

B) We also acquired additional images of some *NG2-Cre^+^/Rosa-DTA* mice brains stained with PDGFRβ and NG2 at E18.5. In all the analyzed brain sections, no decrease in density of PDGFRβ^+^ pericytes could be seen in the dorsal telencephalon. The number of PDGFRβ^+^ pericytes observed in *NG2-Cre^+^/Rosa-DTA* mice was similar to those seen in *NG2-Cre^-^/Rosa-DTA* mice control brains, as also revealed by the quantitative analyses now done by us. Please see the quantitative results in Figure 3 of the revised manuscript. Thus, these additional results confirm that due to the lack of sufficient Cre-mediated recombination in the pericytes, the blood vessel network seen with *NG2-Cre^+^/Rosa-DTA* mice dorsal telencephalon can be mainly attributed to the NG2^+^ glia.

C) It is notable that Zhu et al., 2008 primarily looked at the recombination efficiency in *NG2-Cre^+^/Rosa-tdTomato* mice at P30 only whereas our studies are centered towards late embryonic (and early postnatal) stages only. This age difference (embryonic versus P30) could be the primary reason why we do not see sufficient recombination in pericytes while authors in Zhu et al., 2008 do see significant recombination in pericytes.

D) Next, we aimed to confirm that the GLAST^+^ astrocytes are not affected in the dorsal telencephalon of *Nkx2.1-Cre^+^/Rosa-DTA* mice brains at E18.5. In *Nkx2.1-Cre^+^/Rosa-YFP* mice, recombination can only been seen primarily in the GLAST^+^ astrocytes of the ventral telencephalon whereas no evident recombination can be seen in those that occupy the dorsal telencephalon (as shown in Figure 2). Thus, we did not expect to see a drastic ablation in GLAST^+^ astrocytes of the dorsal telencephalon with *Nkx2.1-Cre^+^/Rosa-DTA* mice brains. Indeed, upon co-immunostaining with GLAST, we found that there was no significant ablation of GLAST^+^ cell population in cingulate cortex of the *Nkx2.1-Cre^+^/Rosa-DTA* mice brains when compared to control *Nkx2.1-Cre^-^/Rosa-DTA* mice (for images and quantification, please see Figure 3—figure supplement 2).

Absence of recombination in the astrocytes of the dorsal telencephalon was the primary reason for deciding to concentrate on the dorsal telencephalon in this manuscript.

Therefore, from the above-mentioned experiments, we are sure that only very few pericytes are affected in *NG2-Cre^+^/Rosa-DTA* mice and no astrocytes are affected in *Nkx2.1-Cre^+^/Rosa-DTA* mice. Moreover, the vascular defects visualized in *NG2-Cre^+^/Rosa-DTA* and *Nkx2.1-Cre^-^/Rosa-DTA* mice brains, are also coherent with our claim that indeed the NG2*^+^* glia are mainly responsible for the observed defects in the blood vessel network. Thus, all the vessel network developmental defects seen with these two mice strains can directly be attributed to primarily NG2*^+^* glia ablation.

2) No mechanistic detail is provided for how NG2 cells carry out their putative vascular regulatory role. It is unclear whether the authors propose that NG2 cells secrete angiogenic factors and how these balance against known roles for neuroepithelial precursors in embryonic brain (e.g., Stenman et al., 2008)? For example, it should be straightforward to assess VEGF or Wnt expression at the appropriate stages of NG2 cell development and then target a loss of function study to show the precise mechanisms involved. Do NG2 cells have direct or indirect effects on vasculature? No angiogenic assays are performed.

We came to the conclusion that the NG2^+^ glia regulate the developing vasculature in the embryonic brains due to their: (i) timely presence in the telencephalon, (iii) close juxtaposition to the developing vessels, and because (iii) we could observe significant vascular defects upon their ablation in the dorsal telencephalon. For the question raised by the reviewers, we had shortly discussed the information about how NG2^+^ glia carry out their regulatory rolein the Discussion section of our previously submitted manuscript and now we have added more details in the revised Discussion section.

The repertoire of factors expressed by NG2^+^ glia themselves are not explored much yet. We are aware that the developing vessel network is dependent on several growth factors such as vascular endothelial growth factor A (VEGF-A), VEGF-B, VEGF-C, VEGF-D and placental growth factor (PlGF) that bind to different tyrosine kinase receptors (VEGFR) expressed on endothelial cells. Other signaling pathways also aid in the development of the vascular circuitry like angiopoietin1/Tie2; Notch/Delta-like-4/JAG1; Wnt signaling and guidance cues with their associated receptors such as ephrin-B2 and EphB4, Slit and Robo4, Netrins and Unc5/Dcc, class 3 semaphorins and Neuropilins/Plexins.

Recently, an elegant study performed in postnatal mice showed that the OPCs-intrinsic Hypoxia-inducible factors 1/2 (HIF1/2) signaling stimulates endothelial cell proliferation in vitro and increases CC white matter angiogenesis in vivo (Yuen et al., 2014). This study leads us to question if the embryonic NG2^+^ glia we have identified also regulate vessels outgrowth through HIF1a and/or HIF2a action. Notably, the population addressed by Yuen et al., 2014 primarily focuses on the effects in postnatal brains, whereas the NG2^+^ glia identified by us disappear by P8. Nonetheless, we proposed that it will be interesting to perform in vivo brain analyses of angiogenesis in mouse embryos from *NG2-Cre* mice crossed with *HIF1a floxed, HIF2a floxed* or *HIF1/2a floxed* mice, allowing selective gene inactivation of HIFs in NG2^+^ glia only.

Several studies have shown that growth factors can regulate the NG2^+^ glia proliferation and migration – of which some belong to PI3K/mTOR and Wnt/β-catenin signaling pathways (Frost et al., 2009; Hill et al., 2013). In particular, Wnt/β-catenin signaling pathway has been suggested to be the critical player required for molecular and cellular steps necessary for proper brain vascularization. As pointed by the reviewers, authors in Stenman et al., 2008 have shown that the broad expression of Wnt7a/7b ligands in the developing early embryonic CNS and induction of Wnt/β-catenin signaling is required for formation and differentiation of the embryonic brain vasculature. The study of Stenman et al., 2008 has also shown that the Wnt/β-catenin signaling is also important for the blood-brain barrier, a structure formed by the brain blood vessels. The effects of the absence of Wnt7a/7b ligands are evident already in early embryonic ages i.e. E12.5 and are crucial for ventral CNS vasculature (Stenman et al., 2008). Notably, the NG2^+^ glia are not present in the dorsal telencephalon as early as E12.5. Hence, a closer dissection of these and other Wnts would be required to ascertain if the NG2^+^ glia are mediating their roles in vascularization through Wnts at later stages i.e. when we observed the defects (E16.5-–E18.5).

Furthermore, several other secreted guidance molecules such as netrin, semaphorins and chemokines have been additionally shown to influence the localization of the NG2^+^ glia (Jarjour et al., 2003; Spassky et al., 2002; Tsai et al., 2002; Piaton et al., 2011; Vora et al., 2012). For a better understanding, a complete dissection of all the different angiogenic factors will be required.

As pointed by the reviewer, the complete understanding of molecular mechanisms undertaken by NG2^+^ glia to guide and aid angiogenesis will provide significant insight into angiogenesis. We aim in the future to identify the exact molecular mechanisms through which NG2^+^ glia act. This would require a complex screening using RT-PCR and extensive histology to verify the mechanisms. Also, as suggested by the reviewers, in vitro and in vivo angiogenic assays will be required for the identification of the angiogenic and growth factors together with testing of anti-angiogenic compounds. Thereafter, the promising candidates will be evaluated in vitroafter siRNA manipulations and ex-vivo using organotypic brain slices (Merz et al., 2013, Merz and Bechmann, 2011). It would be also necessary to analyze the coordinated and oriented cell movements that are necessary for the development of the vascular network especially during sprouting and subsequent maturation phases by performing confocal time-lapse video microscopy studies on embryonic and post-natal organotypic brain slices (Larin et al., 2009). Similar video microscopy techniques allowed us previously to study growth cone behaviors of Tomato-labeled axons while they encountered guidepost neurons along the corpus callosum tract (Niquille et al., 2013). We only aim to perform the aforementioned studies in near future (see the Discussion section).

3) The authors show only one time point for their vascular measurements. Is E18.5 the earliest time point at which they see a difference in the vasculature? Was there a decrease in polydendrocyte numbers prior to that? Do the vascular defects eventually correct themselves after prolonged absence of NG2^+^ glia? A finer parallel time course analysis of the loss of polydendrocytes and the changes in the vasculature should be performed to confirm that the latter follow the former and are not due to non-polydendrocyte-mediated effects of the ablation.

Analyses of vascular network in E14.5 *Nkx2.1-Cre^+^/Rosa-DTA* mice did not show any vessel network formation defects (as shown in Figure 5—figure supplement 1). In *Nkx2.1-Cre^+^/Rosa-DTA* mice, the vascular defects started at E16.5 (as shown in Figure 5—figure supplement 1) and were clearly evident and significant by E18.5 (Figure 4, Figure 5, and 6). Also, longitudinal analyses of NG2^+^ glia in *Nkx2.1-Cre^+^/Rosa-DTA* mouse brains at E14.5, E16.5 and E18.5 was performed by us (see new Figure 4—figure supplement 1) and we saw a drastic decrease in number of NG2^+^ glia in the dorsal telencephalon at E16.5 and E18.5 (in new Figure 4—figure supplement 1 compare panels B to E and C to F).

Importantly, the loss of NG2^+^ glia from the dorsal telencephalon precedes the vessel network defects and therefore, supports their active and crucial role in brain vascularization.

The Nkx2.1-derived NG2^+^ glia cease to exist in later postnatal stages and analyses of postnatal *NG2-Cre^+^/Rosa-DTA* brains at P17-P21 did not reveal any significant blood vessel network defects when compared to littermate *NG2-Cre^-^/Rosa-DTA* brains (see Figure 5—figure supplement 4). Also, the mice do not show any other phenotypic defects.

Thus, it appears that the contribution of the NG2^+^ glia is important to keep up with the fast pace of embryonic brain development, however, postnatally their loss in developing brain is maybe compensated by either later appearing waves of NG2^+^ glia (like shown previously in Kessaris et al., 2006) or through other cells.

We wanted to additionally analyze the *Nkx2.1-Cre^+^/Rosa-DTA* mice at postnatal stages but it was experimentally impossible since these mice die at birth.

4) The introduction of NG2 cells as "polydendrocytes" is confusing for the general reader. Are the authors referring to OPCs or some other type of NG2 glia? What distinguishes a polydendrocyte from an OPC since both do not only make oligodendrocytes? Until the Results section, the description of NG2-labeled cells fails to mention pericytes. Thus, the description of NG2 cells is somewhat confusing and it becomes important in the paper.

We apologize for the confusion generated due to the terminology used by us for the NG2^+^ cells in the submitted manuscript. The cells described in our study have been discriminated from the other cells primarily through the use of antibody against NG2 and *NG2-Cre*^+^ transgenic mice. Other mural cells (especially pericytes) are also labeled by anti-NG2 but are not affected in our ablation studies using *NG2-Cre^+^/Rosa-DTA* mice due to insufficient Cre-mediated recombination in them. Thus, the proposed function of vascular network development in the manuscript is majorly dependent on the presence of NG2^+^ glia.

We do not aim to discriminate the Nkx2.1-derived NG2^+^ glial population from previously identified oligodendrocyte precursor cell (OPC) populations. Though, the Nkx2.1-derived NG2^+^ glial population specified in this study corresponds to the previously identified OPCs (Kessaris et al., 2006), we preferred not to call them as OPCs due to their transient nature and rapid disappearance by P8. Upon postnatal investigation of *Nkx2.1-Cre^+^/Rosa-YFP* brains at P8, we observed that we did not see any substantial number of YFP^+^ cells that were also Olig2^+^ – only rare cases were seen in the dorsal telencephalic region (see new Figure 2—figure supplement 2). Hence, it appears that the Nkx2.1-derived NG2^+^ glial population disappears without giving rise to several oligodendrocytes.

Therefore, in the revised manuscript, we chose to refer cells as “NG2^+^ cells” when we mention the total NG2^+^ profile i.e. polydendrocytes and other mural cells like pericytes and endothelial cells identified by the NG2 antibody.

And, we refer cells as “*NG2^+^ glia*” that are identified by *both* anti-NG2 and transgenic lines (ex: *NG2-Cre*^+^) used in the study – thus, this would exclude the other mural cells leaving behind only NG2^+^ glia or polydendrocytes (also described in previous reports like Nishiyama et al., 2009).

5) The authors state they identify the origins of NG2 glia in the ventral telencephalon. However, the Richardson lab has described this for OPCs in prior studies, importantly Kasaris et al. 2006 Nature Neuroscience, which is not cited. Because NG2 is expressed on OPCs and given findings referred to above, data in Figure 2 seem redundant with prior published work. Indeed, the Kessaris paper showed that early wave OPCs from ventral Nkx2.1^+^ telecephalic precursors are replaced by later waves of OPCs from more dorsal domains. These points and the novelty of the findings need to be clarified.

We apologize that the text created some confusion about the novelty of the data presented in Figure 2. In our submitted version of the manuscript, we had mentioned the publication by Kessaris et al., 2006 with text stating that, “Since *Nkx2.1*-regulated precursors have been shown in embryos to produce transient OPCs in addition to giving rise to GABAergic interneurons and astrocytes, we decided to make use of the *Nkx2.1-Cre^+^/Rosa-YFP* mice”. We realize that we should have added more sentences to explicitly mention previous work done by Kessaris et al., 2006.

With Figure 2, we wanted not only to show the results that have been previously obtained by Kessaris et al., 2006 but we also wanted to show (i) the Cre-mediated recombination efficiency, (ii) restricted temporal placement of the visualized NG2^+^ glia, and (iii) the spatial localization of the NG2^+^ glia in the dorsal telencephalon with our used mice strain. We further wished to highlight the presence and influence of the NG2^+^ glia on vascularization at the time of blood vessel network development that is directly linked to their function in the developing embryonic blood vessel network. There is only a subtle difference between the day of disappearance of NG2^+^ glia observed by us *i. e. P8* and that mentioned in Kessaris et al., 2006 *i.e. P10*. Apart from this, our results are similar to those published in Kessaris et al., 2006.

In order to avoid any confusion about novelty, we have modified the text related to Figure 2 in the Results section, to emphasize the work done by Kessaris et al, 2006 and its direct relevance to our work.